# The Hippo kinase cascade regulates a contractile cell behavior and cell density in a close unicellular relative of animals

**Jonathan E Phillips\*, Duojia Pan\***

Department of Physiology, Howard Hughes Medical Institute, University of Texas Southwestern Medical Center, Dallas, United States

**\*For correspondence:**
jonathan.phillips@
utsouthwestern.edu (JEP);
Duojia.Pan@UTSouthwestern.
edu (DP)

**Competing interest:** The authors declare that no competing interests exist.

**Abstract** The genomes of close unicellular relatives of animals encode orthologs of many genes that regulate animal development. However, little is known about the function of such genes in unicellular organisms or the evolutionary process by which these genes came to function in multicellular development. The Hippo pathway, which regulates cell proliferation and tissue size in animals, is present in some of the closest unicellular relatives of animals, including the amoeboid organism *Capsaspora owczarzaki*. We previously showed that the *Capsaspora* ortholog of the Hippo pathway nuclear effector Yorkie/YAP/TAZ (coYki) regulates actin dynamics and the three-dimensional morphology of *Capsaspora* cell aggregates, but is dispensable for cell proliferation control (Phillips et al., 2022). However, the function of upstream Hippo pathway components, and whether and how they regulate coYki in *Capsaspora*, remained unknown. Here, we analyze the function of the upstream Hippo pathway kinases coHpo and coWts in *Capsaspora* by generating mutant lines for each gene. Loss of either kinase results in increased nuclear localization of coYki, indicating an ancient, premetazoan origin of this Hippo pathway regulatory mechanism. Strikingly, we find that loss of either kinase causes a contractile cell behavior and increased density of cell packing within *Capsaspora* aggregates. We further show that this increased cell density is not due to differences in proliferation, but rather actomyosin-dependent changes in the multicellular architecture of aggregates. Given its well-established role in cell density-regulated proliferation in animals, the increased density of cell packing in *coHpo* and *coWts* mutants suggests a shared and possibly ancient and conserved function of the Hippo pathway in cell density control. Together, these results implicate cytoskeletal regulation but not proliferation as an ancestral function of the Hippo pathway kinase cascade and uncover a novel role for Hippo signaling in regulating cell density in a proliferation-independent manner.

## eLife assessment

This **important** study examines the ancestral function of Hippo pathway kinases in contractility and cell density in the ameboid organism *Capsaspora owczarzaki*, a unicellular animal that is a close relative of multicellular animals. There is **convincing** evidence for Hippo kinases regulating contractility and cell density but not proliferation in *C. owczarzaki*. The work complements previous work on the Hippo effector Yorkie homolog in this species, although the unavailability of extensive genetic tools in this species precludes informative epistasis experiments. The work would be of interest to evolutionary and developmental biologists.

## Introduction

Despite the increasing recognition that the last unicellular ancestor of animals possessed many genetic components that mediate development in modern animals (*Ros-Rocher et al., 2021*), little is known about the function of such developmental regulators in the unicellular ancestor, and the process of how such genes came to function in these roles over evolutionary time is poorly understood. One example of a developmental pathway that is now known to be conserved in close unicellular relatives of animals is the Hippo pathway, which regulates tissue size in animals (*Pan, 2022*). The Hippo pathway consists of a kinase cascade in which the kinase Hippo/MST phosphorylates the kinase Warts/LATS, which in turn phosphorylates the transcriptional coactivator Yorkie/YAP/TAZ, resulting in inactivation through cytoplasmic sequestration (*Figure 1A*, *Ma et al., 2019*). In the nucleus, Yorkie/YAP/TAZ forms a complex with the transcription factor Sd/TEAD, and this complex drives the expression of genes that promote proliferation and affect cell differentiation (*Wu et al., 2008*). The Hippo pathway thus serves to inhibit Yorkie/YAP/TAZ activity, and this regulatory activity restricts tissue growth in development and homeostasis. In cultured cells, the Hippo pathway is regulated by cell density in a process called contact inhibition, whereby Hippo signaling is low in sparse culture to allow cell proliferation but activated in confluent culture to arrest proliferation (*Zhao et al., 2007*). Although the state of the cytoskeleton (*Dupont et al., 2011*) and epithelial cell architecture (*Yang et al., 2015*) are known to regulate Hippo signaling, and many additional upstream regulatory components have been identified (*Deng et al., 2015*; *Hamaratoglu et al., 2006*; *Yu et al., 2010*; *Zheng et al., 2017*), how exactly Hippo signaling is regulated to achieve appropriate tissue size during development or stop cell proliferation in confluent cultures remains incompletely understood. As Hippo pathway dysregulation underlies multiple human cancer types (*Zanconato et al., 2016*), a more complete understanding of Hippo pathway regulation could guide the development of novel cancer therapeutics.

Although once thought to be present only in animals, the Hippo pathway is now known to be present in some of the closest unicellular relatives of animals (*Figure 1B*, *Sebé-Pedrós et al., 2012*). One such unicellular organism is *Capsaspora owczarzaki* from the clade Filasterea, the sister group to Choanozoa (a clade comprising animals and choanoflagellates) (*Shalchian-Tabrizi et al., 2008*). The *Capsaspora* genome encodes all components of the core Hippo pathway as well as some conserved upstream Hippo pathway regulators such as Kibra and NF2/Merlin (*Sebé-Pedrós et al., 2012*). *Capsaspora* is a motile amoeboid organism with a 5-micron-diameter cell body covered with F-actin-rich filopodia-like structures, and was initially identified as an apparent endosymbiont in the snail *B. glabrata* (*Stibbs et al., 1979*). However, all other known filasterians are apparent free-living heterotrophs (*Hehenberger et al., 2017*; *Patterson et al., 1993*; *Tong, 1997*), and whether *Capsaspora* is an obligate or facultative endosymbiont is not known. Physical or chemical signals can induce *Capsaspora* cells to aggregate into round, 3-dimensional multicellular structures that require calcium for cell-cell adhesion (*Ros-Rocher et al., 2023*; *Sebé-Pedrós et al., 2013*). Other filasterians and closely related organisms isolated from natural environments also form aggregates (*Hehenberger et al., 2017*; *Tikhonenkov et al., 2020*), indicating that aggregation is a conserved physiological process in this clade. *Capsaspora* is easily culturable in axenic medium, has a sequenced genome with chromosome-level assembly (*Schultz et al., 2023*; *Suga et al., 2013*), and is genetically tractable owing to gene disruption and stable transgenic techniques that we have recently developed (*Phillips et al., 2022*). Furthermore, the *Capsaspora* genome contains many genes once thought specific to animals, such as cell-cell adhesion and ECM proteins (cadherins, integrins, and laminins) and developmental regulators (NF-κB, P53/63/73, Brachyury, and RUNX) (*Suga et al., 2013*). Therefore, along with other emerging holozoan research organisms (*Booth and King, 2020*; *Faktorová et al., 2020*; *Kożyczkowska et al., 2021*; *Olivetta and Dudin, 2023*; *Woznica et al., 2021*), *Capsaspora* is a valuable system for studying the function of such genes in unicellular organisms.

Previously, we have studied the role of the Hippo pathway in *Capsaspora* by generating a mutant of the single Yorkie/YAP/TAZ ortholog found in the genome (coYki) (*Phillips et al., 2022*). We found that coYki affects cytoskeletal dynamics, with coYki mutants showing constant ectopic blebbing at the cell cortex. Furthermore, coYki affects the morphology of multicellular aggregates: whereas WT aggregates are spheroid, coYki mutant aggregates are flat and less circular than WT. Despite the well-characterized role of Yorkie/YAP/TAZ in promoting proliferation in animals, we saw no effect on cell proliferation in the coYki mutant, suggesting that regulation of proliferation by Yorkie/YAP/TAZ emerged after the last common ancestor of *Capsaspora* and animals. However, two important

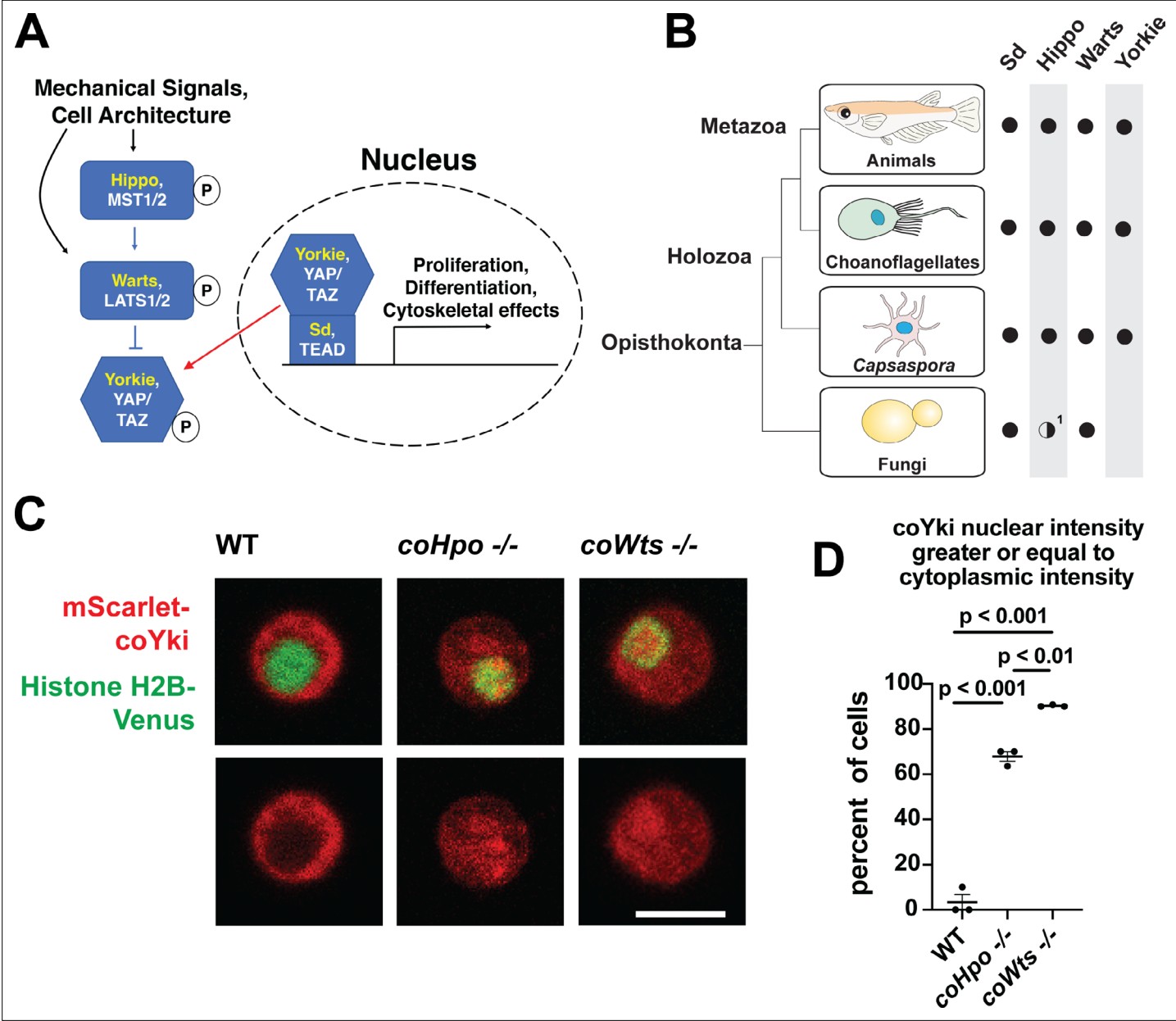

**Figure 1.** The Hippo pathway kinases regulate the subcellular localization of coYki. (**A**) The Hippo pathway in *Drosophila* (yellow text) and mammals (white text). Figure from *Phillips et al., 2022*. (**B**) Conservation of Hippo pathway components in close unicellular relatives of animals. [1]The Hippo kinase SARAH domain is absent from putative Hippo kinase orthologs in yeasts, but is present in the early-branching fungus *Spizellomyces punctatus* and in amoebozoans such as *Dictyostelium*, indicating SARAH domain loss in some fungal lineages. Data from *Sebé-Pedrós et al., 2012*. (**C**) Cells were transiently transfected with the indicated constructs and imaged by confocal microscopy. Scale bar is 5 microns. (**D**) Intensity of mScarlet-coYki signal in the nucleus and cytoplasm of cells was quantified. Where p-values are given, differences between conditions are significant (One-way ANOVA, Tukey's test, n=3 with at least 10 cells measured for each independent experiment.) Values are mean ± SEM. Absence of error bars indicates that error is smaller than the plot symbol.

The online version of this article includes the following source data and figure supplement(s) for figure 1:

**Figure supplement 1.** Knockout of *coHpo and coWts*.

**Figure supplement 1—source data 1.** Original unedited file for the DNA gel image used to generate panel B.

**Figure supplement 1—source data 2.** Uncropped file showing DNA gel image used to generate panel B including annotations of bands and sample labels from panel B.

**Figure supplement 1—source data 3.** Original unedited file for the DNA gel image used to generate panel D.

**Figure supplement 1—source data 4.** Uncropped file showing DNA gel image used to generate panel B including annotations of bands and sample labels from panel D.

questions remain unanswered: (1) do the upstream components of the Hippo pathway regulate coYki, as they do in animals? (2) What effect does an increase of coYki activity have on *Capsaspora*?

In this study, we addressed these questions by generating homozygous mutant cell lines for the Hippo (coHpo) and Warts (coWts) kinases in *Capsaspora*. Loss of either kinase results in increased nuclear localization of coYki, showing that the regulatory activity of the Hippo kinase cascade is conserved. Loss of coHpo or coWts does not increase cell proliferation, consistent with our previous conclusion that the Hippo pathway does not significantly affect proliferation in *Capsaspora*. However, loss of either kinase results in a distinct cytoskeletal phenotype: cells display a contractile behavior in which an initially round cell will elongate on an axis, adhere to the substrate at two terminal adhesion sites connected by an F-actin fiber, and then undergo contraction, returning to a round morphology. Furthermore, we find that loss of these kinases increases the density of cell packing within multicellular aggregates, and we present evidence that this cell packing phenotype involves actomyosin-mediated contractile forces. Transgenic expression of a coYki mutant predicted to be hyperactive results in phenotypes like those observed in the coHpo and coWts mutants, suggesting that these phenotypes are caused by increased coYki activity due to a lack of kinase activity. Together, these results support our previous conclusions that the Hippo pathway regulates cytoskeletal dynamics but not proliferation in *Capsaspora*. In light of its well-established role in cell density-regulated proliferation control in animals, we speculate that the Hippo pathway may represent an ancient mechanism of cell density control preceding the emergence of animal multicellularity.

## Results

### The Hippo pathway kinases regulate the subcellular localization of coYki in *Capsaspora*

Characterizing the function of the Hippo pathway in close unicellular relatives of animals may (a) enhance our understanding of the evolutionary origins and ancestral function of this pathway and (b) provide a relatively simple system to better understand how this biomedically significant pathway is regulated. To address these aims, we generated homozygous mutant cell lines for the single *Capsaspora* orthologs of the Hippo/MST and Warts/LATS kinases (coHpo and coWts, respectively.) For each gene, homologous recombination was used to replace the kinase domain of both WT alleles with selectable markers (Figure S1) as described previously (*Phillips et al., 2022*). Homozygous mutation of the kinases was verified by PCR and by sequencing the disrupted region of the genes (*Figure 1—figure supplement 1*).

In animals, the Hippo kinase cascade regulates the transcriptional coactivator Yorkie/YAP/TAZ by phosphorylation-mediated cytoplasmic sequestration (*Dong et al., 2007*). To test if the Hippo kinase cascade similarly regulates the subcellular localization of the single YAP/TAZ/Yorkie orthologue in *Capsaspora* (coYki), we transiently co-transfected WT, *coHpo-/-*, and *coWts-/-* cells with the fluorescent fusion proteins mScarlet-coYki and Histone H2B-Venus, a marker of the nucleus. In WT cells, mScarlet-coYki was predominantly cytoplasmic (*Figure 1C*), as reported previously (*Phillips et al., 2022*). In contrast, the majority of *coHpo-/-* and *coWts-/-* cells showed nuclear levels of mScarlet-coYki greater than or equal to that seen in the cytoplasm (*Figure 1C and D*). Interestingly, we found that *coWts-/-* cells were significantly more likely to show nuclear mScarlet-coYki localization than *coHpo-/-* cells (*Figure 1D*), which is consistent with Hpo/MST-independent activity of Wts/LATS previously reported in *Drosophila* and mammals (*Zheng et al., 2015*). Together, these results show that the *Capsaspora* Hippo kinase cascade regulates coYki by cytoplasmic sequestration, indicating a premetazoan origin of this regulatory activity of the Hippo pathway.

### The Hippo pathway kinases do not restrict cell proliferation in *Capsaspora*

The Hippo pathway negatively regulates tissue size and cell proliferation by restricting the activity of the growth-promoting oncogene Yorkie/YAP/TAZ in animals. However, we have previously reported that *coYki* is genetically dispensable for cell proliferation in *Capsaspora* (*Phillips et al., 2022*). To further evaluate the role of proliferation control by the Hippo pathway in *Capsaspora*, we examined the requirement of *coHpo* and *coWts* in cell proliferation control under three distinct conditions: adherent growth while attached to a solid substrate, proliferation in shaking culture, and proliferation

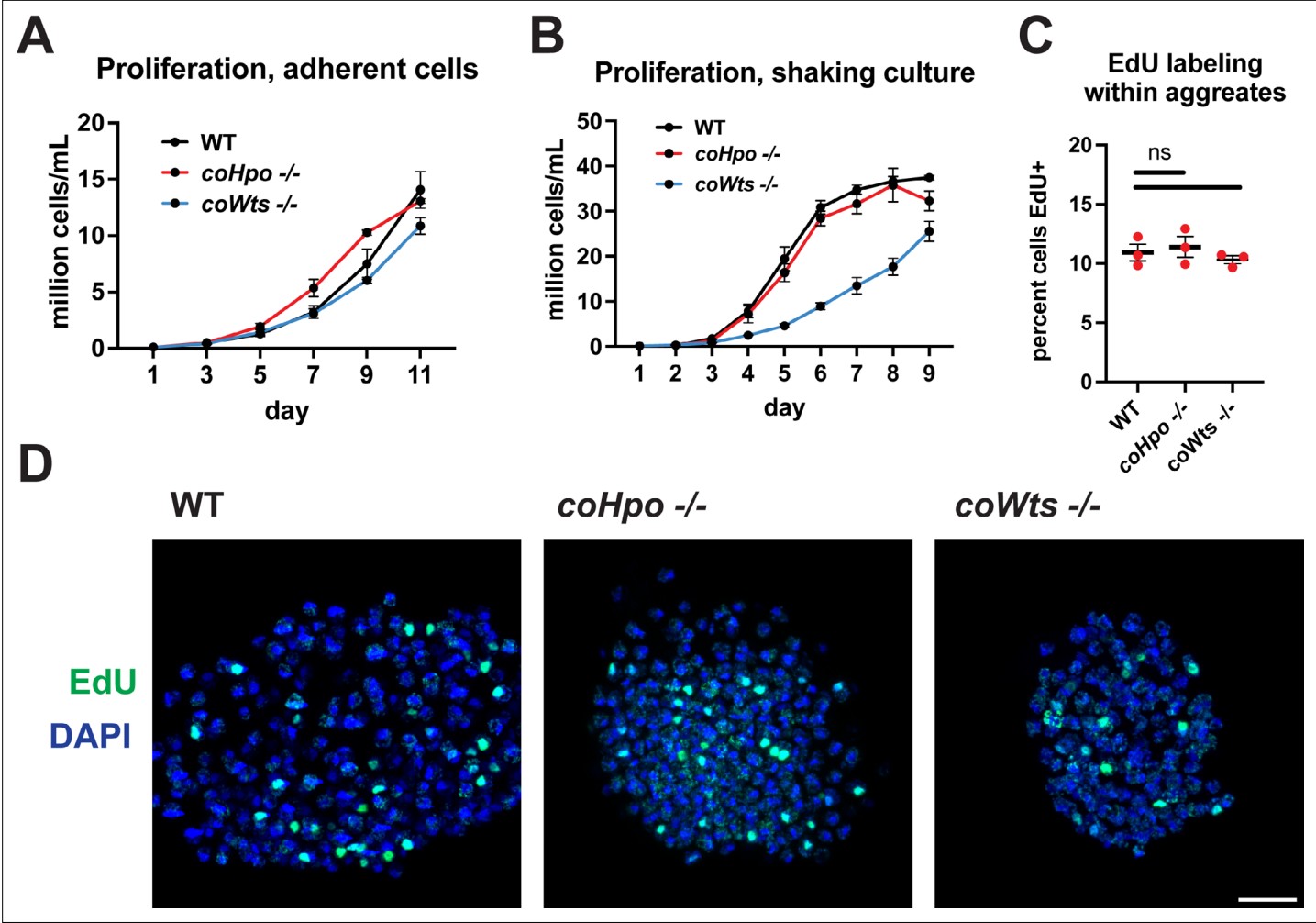

**Figure 2.** The Hippo pathway kinases do not restrict cell proliferation in *Capsaspora*. Cells were grown in adherent culture in 96-well plates (**A**) or in shaking culture (**B**) and cell density was measured daily by hemocytometer. (**C, D**) Cells were inoculated under low-adherence conditions to induce aggregate formation, and 3 days after induction aggregates were incubated with EdU for 4 hr, fixed, and processed to detect EdU labeling. Scale bar is 10 microns. Differences in EdU labeling between WT and mutants were not significant (one-way ANOVA, Dunnett's test, n=3 with three aggregates measured for each independent experiment). Black bars show mean ± SEM, and red dots show values for individual aggregates.

within multicellular aggregates. In adherent growth conditions, *coWts-/-* and *coHpo-/-* cells proliferated at similar rates as WT cells (*Figure 2A*). In shaking culture, *coHpo-/-* cells proliferated at similar rates as WT cells, whereas *coWts-/-* cells proliferated slower than WT (*Figure 2B*). As cells lacking a negative regulator of proliferation are expected to proliferate faster than WT cells, this result suggests that coWts is not a negative regulator of cell proliferation as in *Drosophila* and mammals, but instead is required for normal cell viability under some growth conditions. Finally, to examine cell proliferation within multicellular aggregates, we used EdU labeling, a marker of cells that have undergone the S phase. The percentage of cells labeled by EdU in *coHpo-/-* and *coWts-/-* aggregates was not significantly different from WT aggregates (*Figure 2C and D*), indicating that coHpo and coWts do not affect cell proliferation within multicellular aggregates. Together with our previous characterization of *coYki* (*Phillips et al., 2022*), these results support the view that the Hippo pathway does not restrict cell proliferation in *Capsaspora* as in *Drosophila* and mammals.

### The Hippo pathway kinases regulate cell morphology and actomyosin-dependent contractile behaviors in *Capsaspora*

We have previously shown that, in contrast to WT *Capsaspora* cells with a round and relatively stable actin cortex, *coYki-/-* cells show aberrant, actin-depleted bleb-like protrusions at the cell cortex,

implicating coYki in the control of cytoskeletal dynamics and cell morphology (*Phillips et al., 2022*). To test whether the Hippo pathway kinases affect the same processes in *Capsaspora,* we examined the morphology of adherent *coHpo-/-* and *coWts-/-* cells. While most cells showed a round cell body morphology (*Figure 3A*), a distinct elongated cell morphology was observed in a subset of *coWts -/-* or *coHpo -/-* cells, which was seen much less frequently in WT cells. Quantification of cellular aspect ratio in cell populations showed a non-gaussian distribution, with a distinct population of high aspect ratio cells in *coHpo* or *coWts* mutants (*Figure 3B*). The presence of elongated cells was rescued by transgenic expression of coHpo or coWts in the respective mutant (*Figure 3A and B*), demonstrating that the elongated cell phenotype was specifically due to the absence of these kinases.

To examine the dynamic properties of these elongated cells, we imaged *coHpo -/-* and *coWts -/-* cells by time-lapse microscopy (*Figure 3C* and *Videos 1–3*). These elongated cells are highly dynamic. A previously round, motile cell starts extending along an axis, the direction of which does not necessarily correspond with the cell's previous direction of movement. The cell then stops extending, remaining for approximately 20 s in its elongated form. Filopodia appear enriched at the poles of elongated cells (*Figure 3C*, 2:00 timepoint). To examine filopodia more carefully, we stably expressed NMM-Venus in cells, which labels the cell membrane and allows for visualization of the filopodia (*Parra-Acero et al., 2018*). Filopodia were enriched at the poles of elongated cells in all genotypes (*Figure 3—figure supplement 1*), suggesting that filopodia may play a role in this elongated morphology. After this elongated form, the cell then contracts: adhesion at one of the poles is apparently lost, leading to a rapid contractile movement toward the other pole and returning the cell to the initial round morphology. Together, these results suggest that *Capsaspora* cells can perform a dynamic contractile behavior that is regulated by the Hippo pathway kinases.

The actomyosin cytoskeletal network mediates amoeboid motion and cytokinesis in unicellular organisms and contractile processes such as muscle contraction in animals (*Brunet and Arendt, 2016*). To examine the behavior of the actomyosin network in contractile *Capsaspora* cells, we stably expressed Lifeact-mScarlet, which labels F-actin in *Capsaspora* cells (*Phillips et al., 2022*). As contractile cells in the *coHpo* mutant background tended to show a more elongated morphology than the *coWts* mutant*,* we focused on the *coHpo* mutant for further analysis. We have previously reported that adherent WT *Capsaspora* cells show F-actin-rich foci near the area of contact with the substrate. In motile cells, these foci do not move relative to the substrate, suggesting that they are points of adhesion (*Phillips et al., 2022*). Similar F-actin-rich foci were observed at or near the polar ends of elongated *coHpo -/-* cells (*Figure 4A*, blue arrowheads). In most elongated cells, an F-actin fiber running parallel to the axis of elongation was observed connecting the two polar foci (*Figure 4A*, yellow arrow; *Videos 4 and 5*). During the contractile process, the cell contracts towards one of the foci, which remains intact, while the other substrate-foci interaction is lost, and the cell resumes a round morphology.

Superficially, the actin fibers observed in elongated *coHpo -/-* cells resemble stress fibers, which are contractile cytoskeletal structures found in animal cells with functions in processes such as extracellular matrix remodeling and cell motility (*Tojkander et al., 2012*). Contractility of stress fibers is generated by the motor protein myosin II (*Chrzanowska-Wodnicka and Burridge, 1996*). If myosin mediates the contractile behavior of *coHpo -/-* cells, then inhibition of myosin activity may slow or prevent the contraction of elongated cells, resulting in more cells with an elongated morphology in a population. To test this prediction, we treated cells with the myosin inhibitor blebbistatin and then examined the occurrence of elongated cell morphology. Blebbistatin treatment increased the occurrence of elongated cell morphology in *coHpo -/-* cell populations (*Figure 4B and C*). Interestingly, despite the rarity of elongated cell morphology in WT cell populations, blebbistatin also resulted in a greatly increased occurrence of this morphology in WT cell populations (*Figure 4B and C*). These results suggest that the contractile behavior of *Capsaspora* cells is mediated by myosin activity.

## The Hippo pathway kinases regulate the density of cell packing in *Capsaspora* aggregates via actomyosin-mediated contractility

*Capsaspora* cells can aggregate into three-dimensional multicellular structures. We have previously identified a critical role for coYki in this process: whereas WT *Capsaspora* cells form round, spheroid aggregates, *coYki-/-* cells form asymmetrical and flattened aggregates (*Phillips et al., 2022*). To test whether the Hippo pathway kinases regulate aggregate morphology, we induced aggregate formation

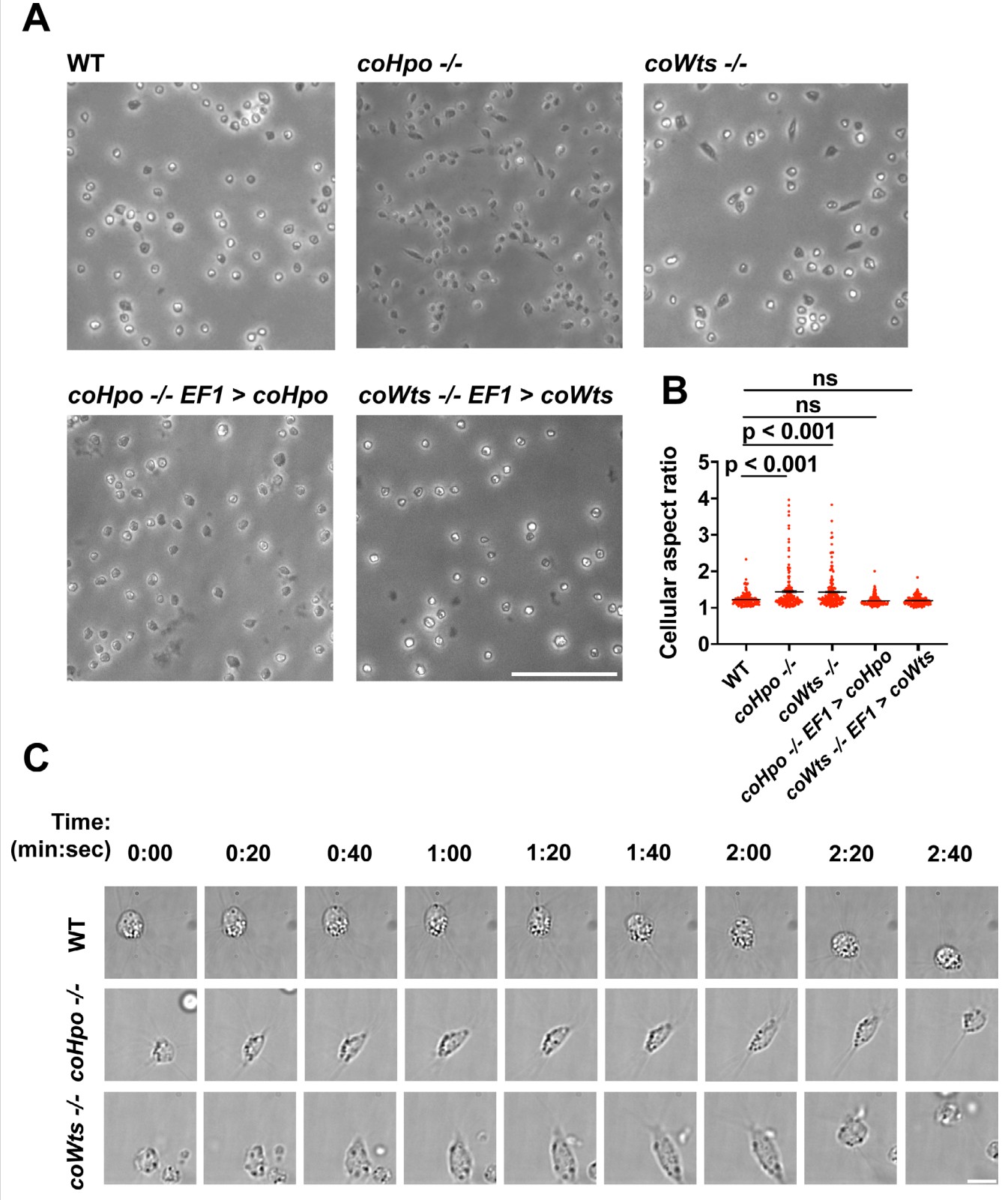

**Figure 3.** The Hippo pathway kinases regulate cell morphology and a contractile cell behavior in *Capsaspora*. (**A**) Cells were inoculated into glass-bottom chamber slides, and adherent cells were imaged 2 days after inoculation. Scale bar is 50 microns. (**B**) The aspect ratio of cells imaged as described in (**A**) was measured using ImageJ. The differences between wild-type (WT) and either *coHpo -/-* or *coWts -/-* cells are significant (one-way

*Figure 3 continued on next page*

*Figure 3 continued*

ANOVA, Dunnett's test, n=3 with 50 cells measured for each independent experiment). Black bars show mean ± SEM, and red dots show values for 150 individual cells. (**C**) Cells were prepared as described in (**A**) and imaged by time-lapse microscopy. Scale bar is 10 microns.

The online version of this article includes the following figure supplement(s) for figure 3:

**Figure supplement 1.** Filopodia are enriched at the poles of elongated cells.

and then measured the size and circularity of aggregates. *coHpo -/-* aggregates showed similar size and circularity as WT aggregates (*Figure 5A and B*). However, *coWts -/-* aggregates, while similar in size to WT, showed reduced circularity. Whereas WT aggregates showed a round, smooth boundary, *coWts -/-* aggregate boundaries were rough and showed nonuniform protrusions (*Figure 5A and B*). These results suggest that Hippo signaling through coWts can affect aggregate morphology, but inactivation of the upstream Hippo kinase alone is not sufficient to induce this effect.

*Capsaspora* aggregates are composed of adherent cells that form connections primarily through filopodia-like protrusions on the cell surface. This can be visualized by mosaic aggregates consisting of cells expressing mScarlet, which labels the cell body, and cells expressing NMM-Venus, which allows for visualization of filopodia. In mosaic aggregates, contacts are visible between filopodia but not between cell bodies (*Figure 5C*), showing that filopodium-filopodium contacts, but not cell body-to-cell body contacts, are the primary mode of adhesion in aggregates.

While examining aggregate morphology, we noticed that *coHpo -/-* aggregates appeared more opaque than WT, suggesting that the Hippo pathway kinases may affect the internal structure of aggregates. To investigate this further, we stably expressed mScarlet in WT, *coHpo -/-*, *coWts -/-*, and *coYki -/-* cells and then used optical sectioning to examine aggregate inner structure. This analysis revealed that cells within *coHpo -/-* or *coWts -/-* aggregates appeared more tightly packed than WT (*Figure 5D*). To quantify this effect, we measured the number of cells per unit area in optical sections of aggregates. Whereas the number of cells per unit area in *coYki -/-* aggregates was not significantly different from WT, the number of cells per area in *coHpo -/-* and *coWts -/-* was significantly increased relative to WT (*Figure 5E*). These results indicate that loss of the Hippo pathway kinases increases cell density within *Capsaspora* aggregates.

Filopodia mediate the adhesion between cells within aggregates (*Figure 5C*). To test whether differences in filopodial morphology may underlie differences in aggregate shape or cell density in the *coHpo* or *coWts* mutants, we generated mosaic aggregates consisting of WT and mutant cells stably expressing distinct fluorescent labels that allow visualization of filopodia. This mosaicism allowed us to examine the filopodia of individual cells within aggregates. Whereas filopodia from *coHpo -/-* cells resembled those of WT and often extended more than a cell diameter distance within the aggregate, filopodia from *coWts -/-* cells were much shorter (*Figure 5—figure supplement 1*). These results suggest that *coWts* affect the morphology of filopodia within aggregates and that differences in filopodia-mediated interactions between cells may underlie the aberrant shape of *coWts -/-* mutant aggregates (*Figure 5A*). However, as both *coHpo*

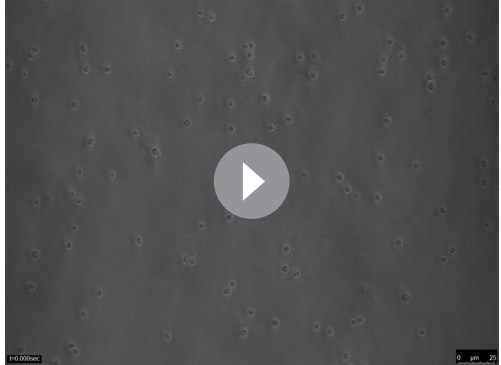

**Video 1.** Time-lapse microscopy of adherent wld-type (WT) *Capsaspora* cells. This video serves as a control for *Videos 2 and 3*.

https://elifesciences.org/articles/90818/figures#video1

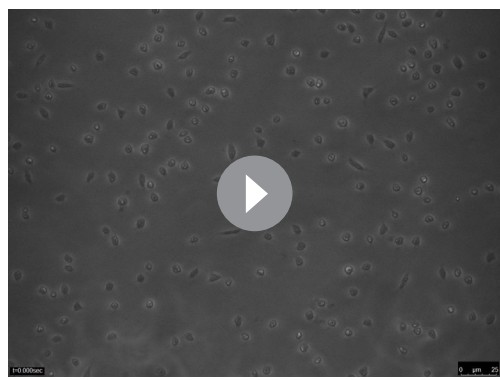

**Video 2.** Time-lapse microscopy of adherent *coHpo -/-* cells.

https://elifesciences.org/articles/90818/figures#video2

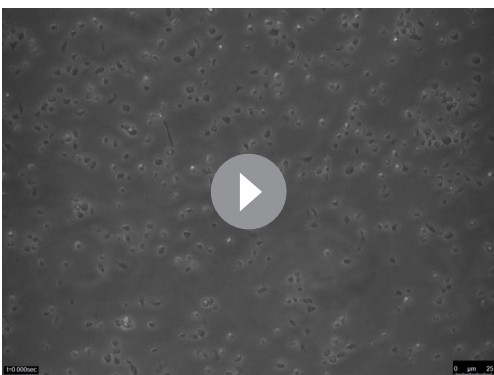

**Video 3.** Time-lapse microscopy of adherent *coWts* -/- cells.

https://elifesciences.org/articles/90818/figures#video3

-/- and coWts -/- aggregates show increased cell density within aggregates, but *coHpo* -/- filopodia do not appear different from WT filopodia, these results suggest that increased cell density is not due to differences in filopodial morphology, but instead may be due to mechanical properties of cells.

Given that *coHpo* and *coWts* regulate a contractile behavior in *Capsaspora* cells, a plausible explanation for the increased cell packing within *coHpo* -/- and *coWts* -/- aggregates is that an increase in intercellular contractile interactions pulls cells closer and, therefore, leads to increased cell density in mutant aggregates. To test whether actomyosin-mediated contractile forces affect cell packing within aggregates, we treated WT and *coHpo* -/- aggregates with the myosin II inhibitor blebbistatin or the actin-depolymerizing drug latrunculin B (LatB). For both WT and *coHpo* -/- aggregates, treatment with either blebbistatin or LatB decreased cell density within aggregates, as reflected by a reduction in the number of cells per unit area (*Figure 6A–D*). Together, these results suggest that the Hippo pathway kinases regulate the density of cell packing in *Capsaspora* aggregates via actomyosin-mediated contractility.

*coYki* affects both the cytoskeletal dynamics of *Capsaspora* cells and the shape of multicellular aggregates (*Phillips et al., 2022*). Given these previous results and the fact that Yorkie/YAP/TAZ is inhibited by the Hippo pathway kinases in other organisms, it is plausible that the phenotypes observed in the *coHpo* and *coWts* mutants are mediated by *coYki*. Since techniques are currently unavailable to create double mutants in *Capsaspora*, we could not test this possibility through classic genetic epistasis. As an alternative approach, we asked whether ectopic activation of *coYki* can phenocopy the loss of *coHpo* or *coWts*, by transgenically expressing a coYki mutant in which the four predicted Wts/LATS phosphorylation sites are mutated to non-phosphorylatable Alanine (coYki 4SA). coYki 4SA shows increased nuclear localization compared to the wild-type coYki protein (*Phillips et al., 2022*), and thus is predicted to be hyperactive. As observed in *coHpo* and *coWts* mutants, stable transgenic expression of coYki 4SA resulted in increased cell packing of *Capsaspora* aggregates (*Figure 7A and B*), as well as an elongated cell morphology in adherent cells (*Figure 7C and D*). These results suggest that the phenotypes observed in *coHpo* and *coWts* mutants are due to coYki activation resulting from a loss of upstream kinase signaling.

Previously, Xu et al., reported that in *Drosophila*, Yorkie can act in a transcription-independent manner at the cell cortex to regulate actomyosin-mediated contractility (*Xu et al., 2018*), raising the possibility that cytoskeletal regulation by coYki is transcription-independent. To test this possibility, we evaluated whether coYki 4SA requires transcriptional activity to affect cell and aggregate morphology. The conserved coYki residue F123 aligns with the human YAP1 residue F95, which is required for YAP-TEAD binding and YAP transcriptional activity (*Li et al., 2010*). The analogous residue in *Drosophila* is not required for the transcription-independent Yorkie activity reported by *Xu et al., 2018*. We found that, in contrast to coYki 4SA, the expression of a coYki 4SA F123A mutant did not affect cell or aggregate morphology (*Figure 7*). This result indicates that these phenotypes are mediated by the transcriptional activity of coYki.

## Effects of the Hippo pathway on gene expression in *Capsaspora*

To better understand how perturbation of Hippo pathway signaling affects gene expression in *Capsaspora*, we performed RNA-seq on *coHpo* -/- and *coWts* -/- cells. We previously identified 15 *Capsaspora* genes differentially expressed in the *coYki* -/- mutant that is predicted to bind actin (*Phillips et al., 2022*). Of these genes, 14 were differentially expressed in the *coHpo* or coWts mutant cell lines, and six of these genes were differentially expressed in both the *coHpo* and *coWts* mutant lines (Differences in gene expression between WT and mutant are significant, with p<0.05 (*Supplementary*

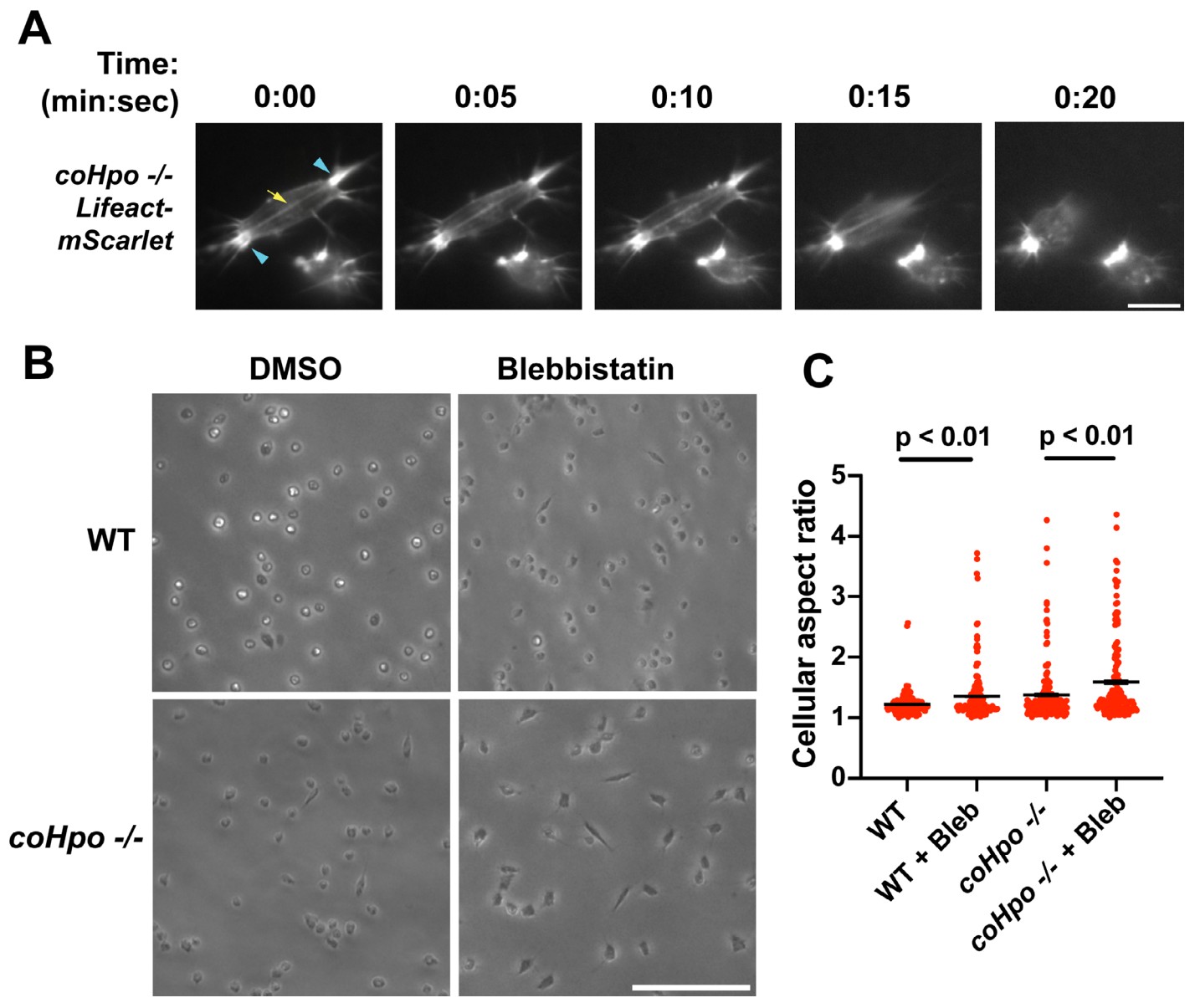

**Figure 4.** Characterization of actomyosin in contractile *Capsaspora* cells. (**A**) Cells were inoculated on glass-bottom chamber slides, and at 2 days after inoculation adherent cells were imaged by time-lapse epifluorescence microscopy. Blue arrowheads indicate ventral F-actin-enriched foci, and the yellow arrow indicates the F-actin fiber that connects the foci. Scale bar is 10 microns. (**B**) Cells were inoculated on glass-bottom chamber slides. Two days after inoculation, DMSO or Blebbistatin at a concentration of 1 μM was added to cells. Cells were imaged 24 hr after the addition of blebbistatin. (**C**) The aspect ratio of cells prepared as in (**B**) was measured using ImageJ. For both wild-type (WT) and *coHpo -/-* cells, the difference between DMSO and blebbistatin treatment measurements are significant (t-test, n=3 with 50 cells measured for each independent experiment). Black bars indicate mean ± SEM, and red dots indicate measurements for 150 individual cells for each condition. Absence of error bars indicates that error is smaller than the plot symbol.

*file 1*)). This result indicates that *coYki*, *coHpo*, and *coWts* regulate a shared set of genes, a subset of which may affect the cytoskeletal properties of cells through actin regulation.

To further examine how the Hippo pathway affects gene expression in *Capsaspora*, we performed functional enrichment analysis on three gene sets: genes differentially expressed in the *coHpo* mutant (1032 genes), the *coWts* mutant (609 genes), and the overlapping set of genes differentially expressed in the *coHpo*, *coWts*, and *coYki* mutants (107 genes, termed the 'core Hippo pathway' gene set) using *coYki* mutant sequencing data from our previous study (*Phillips et al., 2022*)(See *Supplementary file 2* for gene lists). The core Hippo pathway set showed enrichment in membrane-associated

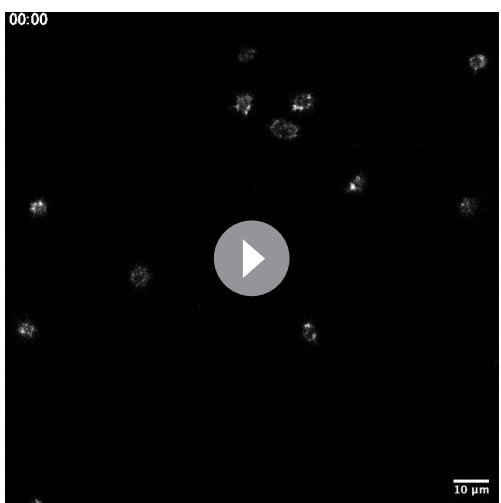

**Video 4.** Time-lapse confocal microscopy of adherent wild-type (WT) cells expressing Lifeact-mScarlet. This video serves as a control for **Video 5**.
https://elifesciences.org/articles/90818/figures#video4

and transmembrane domain-containing proteins, proteins functionally linked to differentiation, and proteins encoding a laminin G domain (**Figure 8**). Laminins are a component of the ECM in animals, functioning to couple cell surface receptors such as integrins to the basement membrane (**Hohenester and Yurchenco, 2013**). The function of Laminin G domain-containing *Capsaspora* proteins is unknown, and conservation of this domain appears restricted to animals and their closest unicellular relatives such as filasterians and choanoflagellates (**Fahey and Degnan, 2012**; **Williams et al., 2014**). Another protein domain, Insulin-like growth factor binding protein, N-terminal (IGFBPN), was also enriched in the core Hippo pathway gene set. In animals, this domain is associated with receptor tyrosine kinases and other proteins linked to tyrosine kinase signaling (**Ward et al., 1995**; **Zesławski et al., 2001**). The sets of genes differentially expressed in the coHpo or coWts mutants also showed enrichment in genes encoding the IGFBPN domain, membrane-associated proteins, and genes functionally linked to differentiation (**Figure 8—figure supplement 1**). The finding that the *Capsaspora* Hippo pathway regulates genes encoding membrane proteins and putative ECM components such as laminins suggests that this pathway may regulate cell and aggregate morphology not just through intracellular cytoskeletal regulators, but also through modification of the extracellular environment and cell surface proteins such as receptors or adhesion molecules.

## Discussion

The evolutionary process by which the signaling pathways that regulate animal development came to function in these roles remains unclear. Apart from shedding light on the origin of animals, knowledge about this process could provide a deeper understanding of these signaling pathways, which play key roles in human health and disease. Although we cannot study the ancestral unicellular organisms that gave rise to animals, studying modern close unicellular relatives of animals can guide our

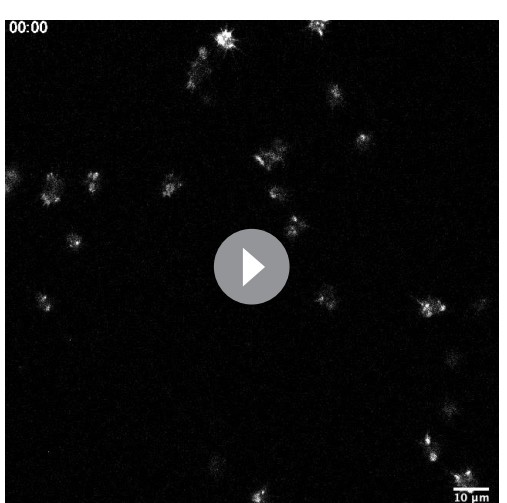

**Video 5.** Time-lapse confocal microscopy of adherent *coHpo -/-* cells expressing Lifeact-mScarlet.
https://elifesciences.org/articles/90818/figures#video5

understanding of what these organisms may have been like, and how conserved signaling pathways may have functioned in these unicellular ancestors. Our previous characterization of the Yorkie/YAP/TAZ ortholog coYki in *Capsaspora* showed that coYki affects actin dynamics and the shape of multicellular aggregates in this close unicellular relative of animals. However, we saw no effect on proliferation in the coYki mutant, suggesting that regulation of proliferation by Yorkie/YAP/TAZ may have evolved later in the lineage leading to animals. In this report, we build on our previous study by assessing the function of the upstream kinases coHpo and coWts in *Capsaspora*. We find that the mechanism of Yorkie/YAP/TAZ regulation (specifically, cytoplasmic sequestration by an upstream kinase cascade consisting of Hippo and Warts kinases) is conserved in *Capsaspora* and animals, indicating an ancient origin of this signaling machinery. These findings support and extend our previous study uncovering a role for

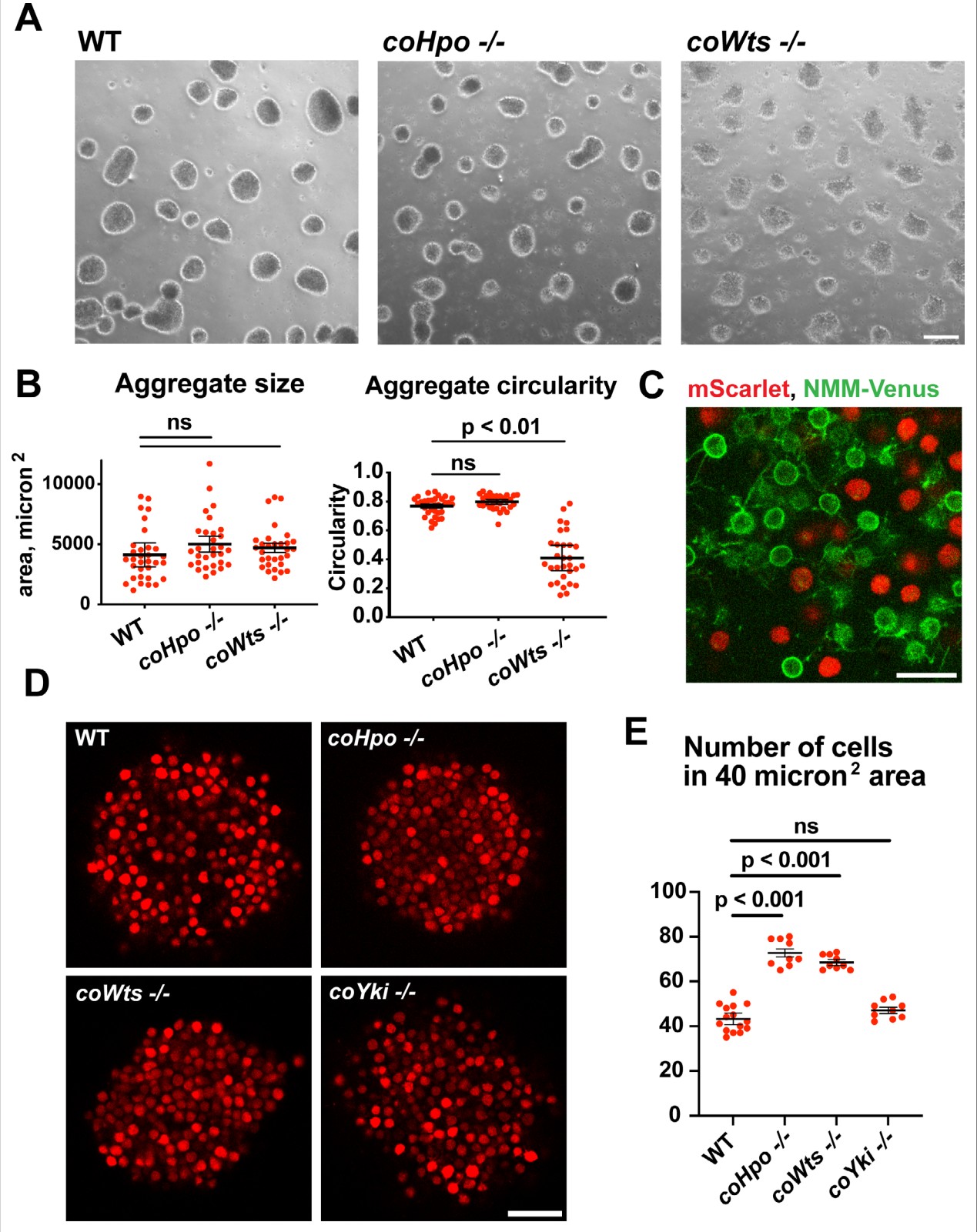

**Figure 5.** The Hippo pathway kinases regulate the density of cell packing in *Capsaspora* aggregates. (**A**) Aggregates were induced by inoculating cells in low-adherence conditions, and aggregates were imaged 5 days after inoculation. Scale bar is 100 microns. (**B**) The size and circularity of aggregates imaged as in (**A**) was measured using ImageJ. Black bars indicate mean ± SEM (n=3 with 10 aggregates measured for each independent experiment), and red dots indicate values for 30 individual aggregates. For aggregate circularity, the difference between wild-type (WT) and *coWts -/-* aggregates

*Figure 5 continued on next page*

*Figure 5 continued*

are significant (one-way ANOVA, Dunnett's test.) (**C**) Cells expressing either mScarlet or N-myristoylation motif (NMM)-Venus, a marker of the plasma membrane which allows visualization of filopodia, were mixed 1:1 and inoculated into low-adherence conditions to induce aggregation. After 5 days, aggregates were imaged by confocal microscopy. Scale bar is 10 microns. (**D**) Clonal populations of cells of the indicated genotype stably expressing mScarlet were inoculated into low-adherence conditions, and cell aggregates were imaged 5 days after inoculation by confocal microscopy. Scale bar is 20 microns. (**E**) The number of cells within a 40 micron$^2$ area from aggregate optical sections imaged as in (**D**) was measured. Black bars indicate mean ± SEM (n=3 with three aggregates measured for each independent experiment), and red dots indicate values for nine individual aggregates for each condition. The difference between WT and either *coHpo -/-* or *coWts -/-* cells is significant (one-way ANOVA, Dunnett's test.).

The online version of this article includes the following figure supplement(s) for figure 5:

**Figure supplement 1.** *coWts* -/- cells within multicellular aggregates show altered filopodial morphology and reduced filopodial length.

Hippo signaling in the regulation of cytoskeletal dynamics but not cell proliferation. Together, our studies suggest that the ancestral function of the Hippo pathway was cytoskeletal regulation, and that subsequently the pathway was co-opted to regulate cell proliferation and tissue size in animals.

In this report, we describe a contractile cell behavior observed in *coHpo* and *coWts* mutants. In this process, the cell transitions from a round morphology to an elongated spindle shape. The cell will remain in the spindle shape for approximately 20 s and then contract towards one pole. We interpret this as a process in which cells extend and then adhere to the substrate at the two ends of the spindle, contractile forces within the cell increase, and adhesion at one of the poles is subsequently lost, leading to contraction towards the other pole. Visualization of actin in live cells is consistent with this model: we observed actin-rich foci at the adhesive sites at the poles of spindle-shaped cells, and an actin filament connecting these foci, which may serve as a tension-generating structure. The cytoskeletal structures that we observe resemble adhesive/contractile structures in mammalian cells such as podosomes (*Linder et al., 2023*), focal adhesions (*Fierro Morales et al., 2022*), and stress fibers (*Tojkander et al., 2012*). As *Capsaspora* encodes orthologs of some proteins known to label these structures, future colocalization experiments may reveal the degree of conservation between such subcellular structures in *Capsaspora* and animals.

The function of these contractile cells is presently unclear. While we observed the contractile behavior on glass surfaces, which is non-physiological, we speculate that such contraction exerted on a pliable surface may remodel a cell's extracellular environment to provide certain fitness benefits. Intriguingly, the elongated, spindle morphology was reported to be common (nearly 50 percent of cells) after the isolation of *Capsaspora* cells from the *B. glabrata* snail host (*Stibbs et al., 1979*), suggesting that environmental factors from the snail host may affect this elongated morphology and potentially Hippo pathway signaling. We further speculate that increased contractility within multicellular aggregates may serve to pull the neighboring cells closer to each other, which may contribute to the increased packing in the *coHpo* or *coWts* mutant aggregates. Given that the elongated cell morphology is rarely but occasionally observed in the WT background, an interesting possibility is that coYki is transiently activated in these cells. Further studies utilizing in vivo reporters of coYki activity may reveal whether this is the case.

We show that the elongated cell morphology visible in *coHpo* and *coWts* mutants becomes more common after treatment with blebbistatin, indicating a role for myosin II in the contraction of this elongated cell state. One interpretation of the increase in elongated cells in the *coHpo* and *coWts* mutants, therefore, is that myosin activity is reduced. However, we also see that blebbistatin treatment reduces the packing of cells within multicellular groups, whereas loss of *coHpo* or *coWts* causes an increase in cell packing. Thus the specific role of myosin activity in the phenotypes observed in Hippo pathway mutants in *Capsaspora* remains unclear. One possibility is that the initial process of cell elongation is unrelated to myosin, and it is this process that is enhanced in *coHpo* and *coWts* mutants. Although techniques to visualize and quantify myosin activity in *Capsaspora* have not been developed, future work in this regard could elucidate the relationship between myosin activity and Hippo signaling in *Capsaspora*.

Previous work has examined the structure of *Capsaspora* aggregates by electron microscopy (*Sebé-Pedrós et al., 2013*). In this report, we have furthered our understanding of aggregate structure by providing evidence that (a) cells within aggregates appear to contact their neighbors largely by interactions between filopodia; and (b) Hippo pathway activity affects the packing of cells within these aggregates. In animals, the predominant form of multicellular structure is epithelia, in which cells with

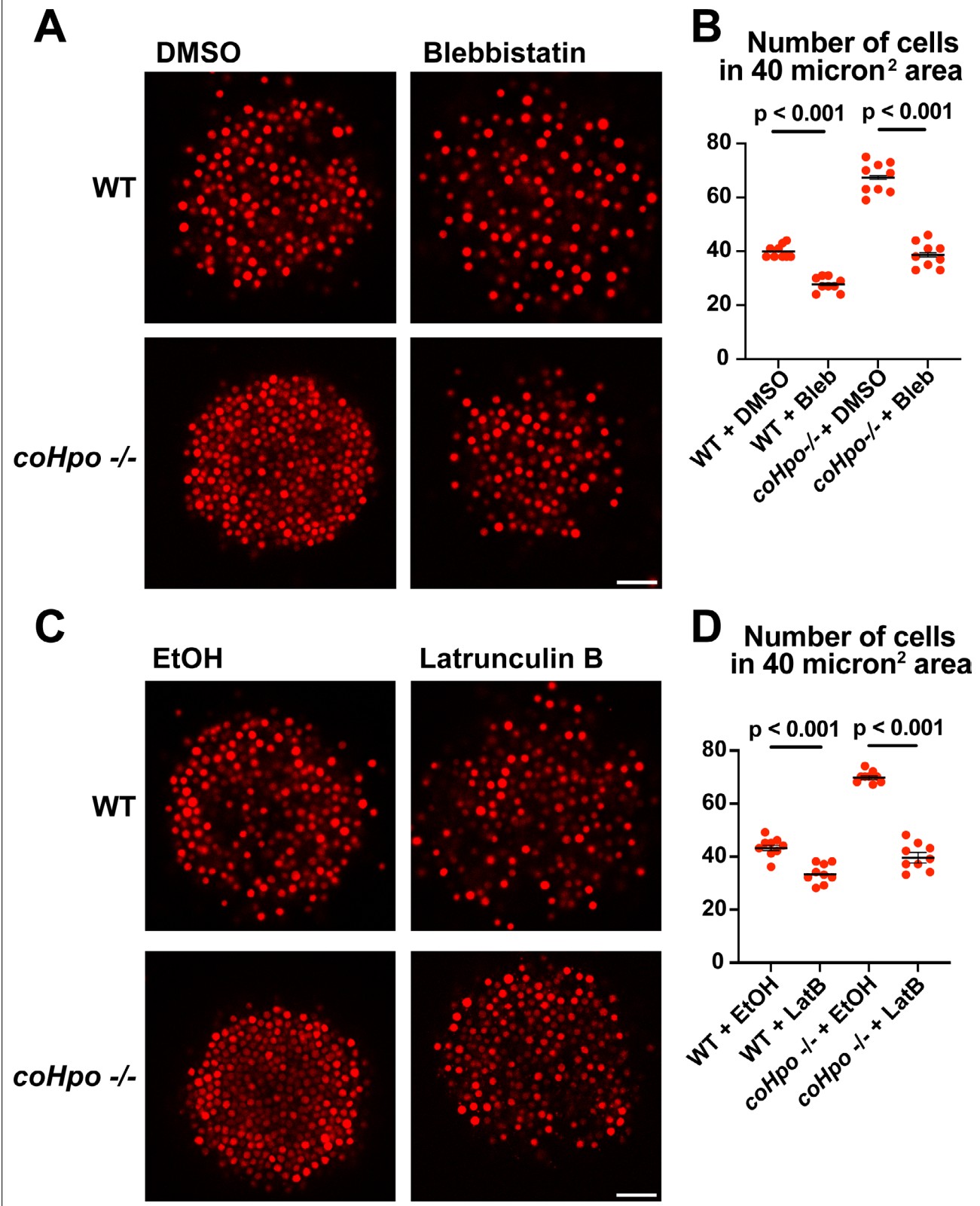

**Figure 6.** Perturbation of the actomyosin network affects cell packing in multicellular *Capsaspora* aggregates. (**A**) Aggregates were treated with 20 µM blebbistatin or DMSO as a vehicle control for 20 minand then imaged by confocal microscopy. Scale bar is 20 microns. (**B**) Cells per area were quantified from images obtained as in (**A**). Black bars indicate means ± SEM (n=3 with three aggregates measured for each independent experiment), and red dots indicate values for nine individual aggregates for each condition. Absence of error bars indicates that error is smaller than the plot symbol. For both

*Figure 6 continued on next page*

Figure 6 continued

wild-type (WT) and *coHpo -/-* aggregates, the difference between DMSO and blebbistatin treatment is significant (t-test). (**C**) Aggregates were treated with 5 µg/ml Latrunculin B or ethanol as a vehicle control for 1 hr and then imaged by confocal microscopy. Scale bar is 20 microns. (**D**) Cells per area were quantified from images obtained as in (**C**). For both WT and *coHpo -/-* aggregates, the difference between ethanol and Latrunculin B treatment is significant (n=3, t-test.).

apical-basal polarity form lateral contacts at adherens junctions, septate junctions, tight junctions, and desmosomes (*Abedin and King, 2010*). However, microvilli/filopodial adhesion plays important roles in dorsal closure in *Drosophila* (*Jacinto et al., 2000*), early embryo compaction in mouse (*Fierro-González et al., 2013*), and other developmental processes (*Brunet and Booth, 2023*). Furthermore, inter-microvillar contacts are the apparent means of adherence in the choanoflagellate *C. flexa*, another close unicellular relative of animals, that can exist in a sheet-like morphology with a remarkable actomyosin-dependent inverting activity (*Brunet et al., 2019*). Thus microvilli/filopodia-mediated

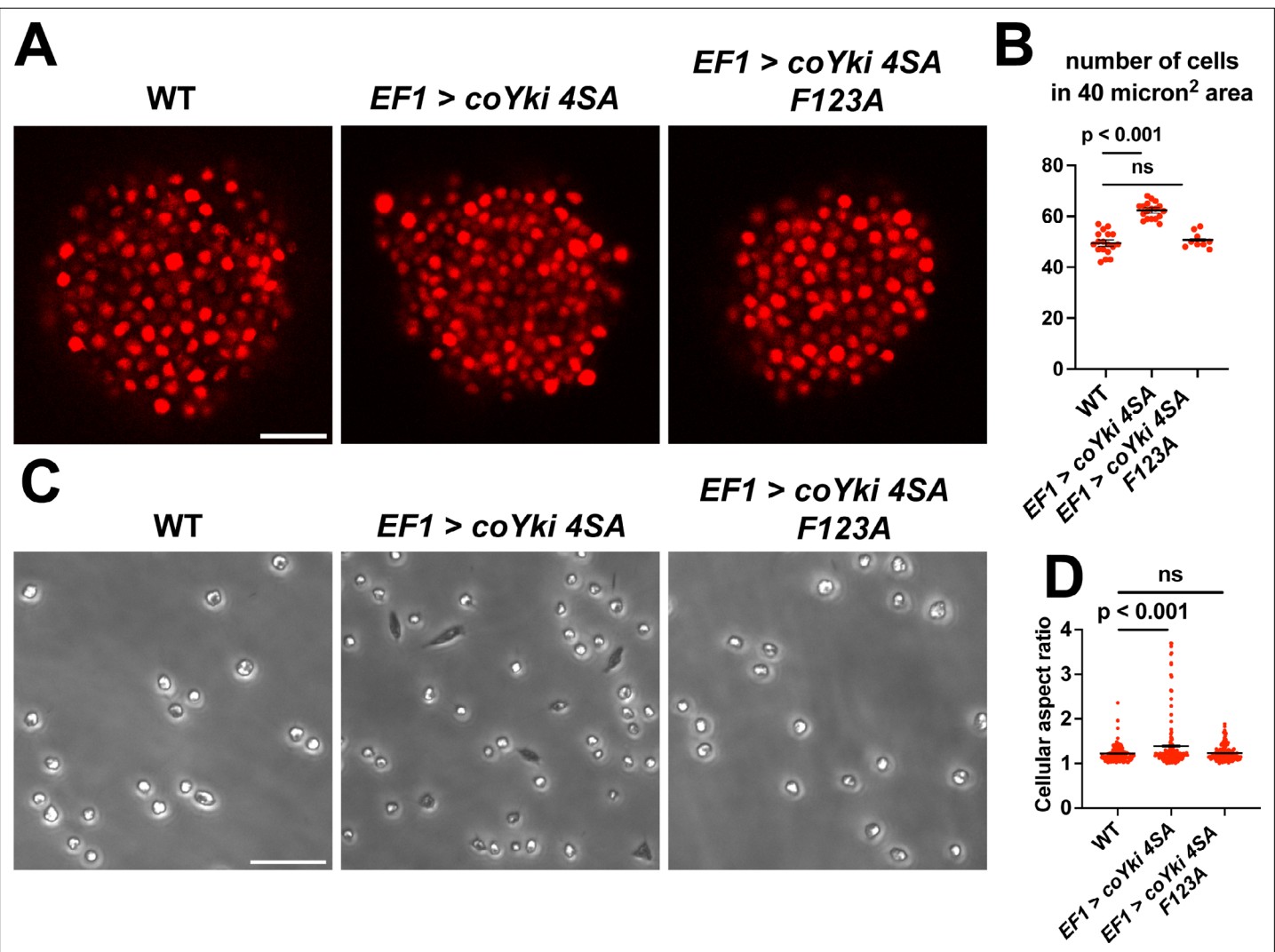

**Figure 7.** Expression of a constitutively active form of coYki phenocopies loss of *coHpo* or *coWts* in *Capsaspora*. (**A**) Aggregates were imaged 5 days after inoculation. Scale bar is 20 microns. (**B**) The number of cells within a 40 micron$^2$ area within an aggregate was measured. Black bars indicate mean ± SEM (n=3 with three aggregates measured for each independent experiment), and red dots indicate values for 9 individual aggregates for each condition. The difference between wild-type (WT) and *EF1 >coYki 4SA* is significant (one-way ANOVA, Dunnett's test). (**C**) Cells were inoculated into glass-bottom chamber slides and imaged 2 days after inoculation. Scale bar is 25 microns. (**D**) The aspect ratio of cells imaged as in (**C**) was measured using ImageJ. Black bars show mean ± SEM (n=3 with 40 cells measured for each independent experiment), and red dots show values for 150 individual cells. The difference between WT and *EF1 >coYki 4SA* is significant (one-way ANOVA, Dunnett's test).

## Enriched categories, set of overlapping genes differentially expressed in *coHpo*, *coWts*, and *coYki* mutants

**Figure 8.** Enrichment analysis of the overlapping set of genes differentially expressed in *coHpo*, *coWts*, and *coYki* mutant cells. Categories with FDR <0.05 are shown. Interpro or Uniprot accession numbers are given for each enriched category. FDR: false discovery rate.

The online version of this article includes the following figure supplement(s) for figure 8:

**Figure supplement 1.** Enrichment analysis for genes differentially expressed in coHpo or coWts mutant cells.

adhesion may be an ancient and conserved means of multicellularity that preceded the formation of epithelia.

In cultured mammalian cells, Hippo signaling regulates contact inhibition, a phenomenon in which proliferation is arrested upon cell confluency to achieve a proper cell density (*Aragona et al., 2013*; *Zhao et al., 2007*). Interestingly, while Hippo signaling does not regulate cell proliferation in *Capsaspora*, loss of Hippo pathway kinases or increase in coYki activity increases the density of cell packing within *Capsaspora* aggregates. Thus, despite the distinct routes of achieving proper cell density (cell proliferation vs. cell aggregation), the Hippo pathway appears to restrict cell density in both animals and *Capsaspora*. We speculate that the Hippo pathway may represent an ancient mechanism of cell density control preceding the emergence of animal multicellularity. Significantly, this model suggests that the signals that regulate the Hippo pathway in animals and *Capsaspora* may be conserved, and therefore studying *Capsaspora* could provide important insights into the regulation of this biomedically significant pathway.

## Materials and methods

**Key resources table**

| Reagent type (species) or resource | Designation | Source or reference | Identifiers | Additional information |
|---|---|---|---|---|
| Gene (*Capsaspora owczarzaki*) | *coHpo* | Genbank | CAOG_01932 | |
| Gene (*Capsaspora owczarzaki*) | *coWts* | Genbank | CAOG_00619 | |

*Continued on next page*

*Continued*

| Reagent type (species) or resource | Designation | Source or reference | Identifiers | Additional information |
|---|---|---|---|---|
| Gene (*Capsaspora owczarzaki*) | *coYki* | Genbank | CAOG_07866 | |
| Cell line (*Capsaspora owczarzaki*) | WT | ATCC | 30864 | |
| Recombinant DNA reagent | pJP80 | *Phillips et al., 2022* | | *Capsaspora* expression vector: mScarlet-coYki fusion, Geneticin resistance |
| Recombinant DNA reagent | pONSY-coH2B:Venus | Addgene | Addgene Plasmid #111877 | *Capsaspora* expression vector: Histone H2B-Venus fusion |
| Recombinant DNA reagent | pJP122 | This paper | | *Capsaspora* expression vector: coHpo transgene, Hygromycin B Resistance |
| Recombinant DNA reagent | pJP125 | This paper | | *Capsaspora* expression vector: coWts transgene, Hygromycin B Resistance |
| Recombinant DNA reagent | pJP145 | This paper | | *Capsaspora* expression vector: NMM-Venus, Hygromycin B Resistance |
| Recombinant DNA reagent | pJP118 | *Phillips et al., 2022* | | *Capsaspora* expression vector: Lifeact-mScarlet, Hygromycin B Resistance |
| Recombinant DNA reagent | pJP103 | *Phillips et al., 2022* | | *Capsaspora* expression vector: mScarlet, Hygromycin B Resistance |
| Recombinant DNA reagent | pJP90 | *Phillips et al., 2022* | | *Capsaspora* expression vector: coYki 4SA, Geneticin Resistance |
| Sequence-based reagent | JP201: coHpo KO Forward: homologous arm +actin promoter | This paper | PCR primers | CCCGCCTCGTCCCAACAACAGCTCGTACGGC TCGGTCTTCAAGGCTCGCCACAAGGACACCCAG TCCATCCTCGCCGTCAAGCAGGTGCCCCTTGAG AAC**ACAAAAATGCTGATTGTTTG** |
| Sequence-based reagent | JP202: coHpo KO Reverse: homologous arm +actin terminator | This paper | PCR primers | AAGCAACATTGCTTCGTTGACTAACGCTTCATGG TGCCGTCCTCGCCCATGACCATCGTGCCGGAG TCGC CAGTGACCATCGTTCCGCTATCAACAAGG G**TTTTTTCTTTGTACAAGATCAC** |
| Sequence-based reagent | JP203: coWts KO Forward: homologous arm +actin promoter | This paper | PCR primers | CAAACGGCGAGACCAGCTCGAGATTGAGATGG CCAAGATGGACCTGACCGATGTTCAAAAGACCC AGTTGCGTCGCATCCTCCGCATGAAGGAATCA GAG**ACAAAAATGCTGATTGTTTG** |
| Sequence-based reagent | JP204: coWts KO Reverse: homologous arm +actin terminator | This paper | PCR primers | GCATTTTAATGTGCTCGACCGTCTCCCGGCCG AGACGATCGGAGGATTCACAGCAGAGCCGGGA GATGAGGTCTTTGGACTCGCGCGAAATCTTTGC CCG**TTTTTTCTTTGTACAAGATCAC** |
| Sequence-based reagent | JP205: coHpo Diagnostic Forward | This paper | PCR primers | CAAAAACAACAGCGAAAACG |
| Sequence-based reagent | JP206: coHpo Diagnostic Reverse | This paper | PCR primers | CCATCGGAGGAAACTAAAGG |
| Sequence-based reagent | JP207: coWts Diagnostic Forward | This paper | PCR primers | TGCTGCTGAAAATGAAAACG |

*Continued*

| Reagent type (species) or resource | Designation | Source or reference | Identifiers | Additional information |
|---|---|---|---|---|
| Sequence-based reagent | JP208: coWts Diagnostic Reverse | This paper | PCR primers | GAACCTCACCAGGATTTTGC |
| Commercial assay or kit | TransIT-X2 Dynamic Delivery System | Mirus Bio | MIR 6003 | |
| Commercial assay or kit | Click-iT Plus EdU Cell Proliferation Kit for Imaging, Alexa Fluor 488 dye | ThermoFisher | C10637 | |
| Chemical compound, drug | Blebbistatin | Sigma-Aldrich | B0560 | |
| Chemical compound, drug | Latrunculin B | Sigma-Aldrich | L5288 | |

## Cell culture

*Capsaspora* cells were grown in ATCC medium 1034 as described previously (*Phillips et al., 2022*). Quantification of proliferation in shaking culture, induction of aggregates, quantification of aggregate size and circularity, EdU labeling within aggregates, and quantification of mScarlet-coYki fluorescence intensity were done as described previously (*Phillips et al., 2022*). Quantification of proliferation on a solid substrate was done as described previously (*Phillips et al., 2022*), except 96-well plates were used in place of 24-well plates. To add blebbistatin (Sigma B0560) or latrunculin B (Sigma L5288) to cell cultures, a 10 X mix of drug or vehicle control in a growth medium was prepared, and this concentrated drug preparation was added to cultures so that a final 1 X concentration of drugs was reached.

## Molecular biology

For a list of plasmids and oligonucleotides generated for this study, see Key Resources Table. For a list of synthesized DNA sequences used in this study, see *Supplementary file 3*. For transgenic expression of *Capsaspora* genes, we synthesized genes coding for the appropriate *Capsaspora* protein but with synonymous codon changes throughout the DNA sequence. This was done to reduce DNA sequence homology in order to prevent homologous recombination of the transgene with the endogenous *Capsaspora* alleles during plasmid integration into the genome, as done previously (*Phillips et al., 2022*). To construct a plasmid for transgenic expression of coHpo in *Capsaspora* cells (pJP122), we synthesized a transgene encoding the coHpo protein (sJP201). There are three predicted isoforms of the coHpo sequence, but only one of these encodes the SARAH domain, which is important for Hippo kinase activity in Hippo pathway signaling (*Scheel and Hofmann, 2003*). We, therefore, used the isoform sequence encoding the SARAH domain to design our transgene. The coHpo transgene was cloned into a KpnI and AflII digest of pJP103 (Addgene #176481) using Gibson assembly, generating a vector encoding the coHpo transgene and hygromycin B resistance. Similarly, a coWts transgene was synthesized (sJP202) and cloned into a KpnI and AflII digest of pJP103 by Gibson assembly, generating a vector encoding a coWts transgene and hygromycin B resistance (pJP125.)

To generate a plasmid to stably express N-myristoylation motif (NMM)-Venus (pJP145), a marker of the plasma membrane and filopodia in *Capsaspora*, we first constructed a vector expressing both Venus and hygromycin B resistance (pJP115) by PCR-amplifying the Venus gene from pJP114 (*Phillips et al., 2022*) and cloning it by Gibson assembly into a KpnI and AflII digest of pJP103 (Addgene #176481), which contains a hygromycin B resistance cassette. We then synthesized the NMM motif (sJP203) following the sequence from the pONSY-coNMM:mCherry vector (Addgene #111878, *Parra-Acero et al., 2018*) and cloned this DNA fragment into a KpnI digest of pJP115 using Gibson assembly.

## *Capsaspora* transfection and genome editing

Transfection of *Capsaspora* cells and stable cell line generation was done using the *Trans*IT-X2 transfection reagent (Mirus Bio) as previously described (*Phillips et al., 2022*). Disruptions of *Capsaspora* genes were done as previously described (*Phillips et al., 2022*). Briefly, oligos oJP201 and oJP202

were used to generate gene-targeting constructs by PCR with homologous arms that target the coHpo gene (CAOG_01932). Oligos oJP203 and oJP204 were used to generate gene-targeting constructs for the coWts gene (CAOG_00619). For both homozygous mutants, the two WT alleles present in the diploid *Capsaspora* genome were disrupted using selectable markers for geneticin resistance and nourseothricin resistance. oJP205 and oJP206 were used in a diagnostic PCR that amplifies across the deleted region of the *coHpo* gene, and oligos oJP207 and oJP208 were used in a diagnostic PCR across the deleted region of the *coWts* gene.

## Measurement of cell and aggregate properties

To measure the aspect ratio of cells, the ImageJ freehand selection tool was used to manually trace the border of imaged cells, and the Measure function was used to calculate the cellular aspect ratio. To quantify cell packing within aggregates expressing mScarlet, a z-stack of optical sections was obtained by confocal microscopy. To measure cells within the aggregate as opposed to cells at the edge of the aggregate, an optical section 14 microns deep into the aggregate was used for measurement. A 40 micron$^2$ square was drawn in the center of the optical section of the aggregate and the number of cells in the square was counted.

## RNA-seq

RNA-seq of *Capsaspora* cells was done as previously described (*Phillips et al., 2022*) except that cells were inoculated into flasks at $4 \times 10^5$ cells/ml instead of $2 \times 10^5$ cells/ml to increase RNA yield. Analysis of RNA-seq data was done as previously described (*Phillips et al., 2022*), except that after differential expression analysis by edgeR (*Robinson et al., 2010*), the following criteria were used to select genes for further analysis: false discovery rate [FDR]≤0.001, absolute log2(fold change)≥0.5, log(counts per million)≥0. Enrichment analyses were done using DAVID (*Sherman et al., 2022*).

## Gene nomenclature

In this article, we use italicized text to refer to a gene (e.g. *coHpo*), unitalicized text to refer to a protein (e.g. coHpo), and '-/-' to indicate a homozygous deletion mutant (e.g. *coHpo -/-*). '*EF1 >coHpo*' indicates transgenic expression of *coHpo* driven by the *EF1* promoter.

## Statistics

Statistical analyses were done using Prism (GraphPad Software, San Diego, CA). For experiments where multiple cells or aggregates were measured for each independent experiment, the mean of all measurements made for each biological replicate was calculated, and statistical analyses were performed on these sample level means following *Lord et al., 2020*.

## Acknowledgements

We thank Maribel Santos for technical assistance with genotyping cell lines. We thank Dr. Iñaki Ruiz-Trillo for the gift of pONSY-coH2B:Venus vector (Addgene plasmid number 111877). This work was supported in part by National Institute of Health grant EY015708 to D.P. During this work JEP was a Howard Hughes Medical Institute fellow of the Life Sciences Research Foundation. DP is an investigator of the Howard Hughes Medical Institute.

## Additional information

### Funding

| Funder | Grant reference number | Author |
| --- | --- | --- |
| Howard Hughes Medical Institute | | Duojia Pan |
| National Eye Institute | EY015708 | Duojia Pan |

| Funder | Grant reference number | Author |
|--------|------------------------|--------|

The funders had no role in study design, data collection and interpretation, or the decision to submit the work for publication.

## Author contributions

Jonathan E Phillips, Conceptualization, Data curation, Formal analysis, Investigation, Methodology, Writing – original draft, Writing – review and editing; Duojia Pan, Conceptualization, Supervision, Funding acquisition, Methodology, Project administration, Writing – review and editing

## Author ORCIDs

Jonathan E Phillips ⓘ https://orcid.org/0000-0001-9896-5895
Duojia Pan ⓘ http://orcid.org/0000-0003-2890-4645

Reviewer #1 (Public Review): https://doi.org/10.7554/eLife.90818.3.sa1
Reviewer #2 (Public Review): https://doi.org/10.7554/eLife.90818.3.sa2
Reviewer #3 (Public Review): https://doi.org/10.7554/eLife.90818.3.sa3
Author Response https://doi.org/10.7554/eLife.90818.3.sa4

# Additional files

## Supplementary files

• Supplementary file 1. Genes with actin-binding functional annotation showing differential expression in *coYki* mutant cells and also exhibiting differential expression in *coHpo* and/or *coWts* mutant cells.

• Supplementary file 2. Lists of genes differentially expressed in *coHpo* mutant cells, coWts mutant cells, and the overlapping set of genes differentially expressed in *coHpo*, *coWts*, and *coYki* mutant cells.

• Supplementary file 3. Sequences of synthesized gene fragments used in this study.

• MDAR checklist

## Data availability

RNA-seq data generated in this study have been deposited into the NCBI SRA (BioProject accession number PRJNA1060651).

The following dataset was generated:

| Author(s) | Year | Dataset title | Dataset URL | Database and Identifier |
|-----------|------|---------------|-------------|-------------------------|
| Phillips JE, Pan D | 2024 | Effect of Hippo and Warts kinases on gene expression in Capsaspora owczarzaki | https://www.ncbi.nlm.nih.gov/bioproject/PRJNA1060651 | NCBI BioProject, PRJNA1060651 |

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
