## [Editor Report · eLife assessment]

This **important** study examines the ancestral function of Hippo pathway kinases in contractility and cell density in the ameboid organism *Capsaspora owczarzaki*, a unicellular animal that is a close relative of multicellular animals. There is **convincing** evidence for Hippo kinases regulating contractility and cell density but not proliferation in *C. owczarzaki*. The work complements previous work on the Hippo effector Yorkie homolog in this species, although the unavailability of extensive genetic tools in this species precludes informative epistasis experiments. The work would be of interest to evolutionary and developmental biologists.

---

## [Referee Report · Reviewer #1 (Public Review)]

Summary:

This Research Advance is an extension of this group's prior eLife paper published in 2022 on the conserved roles of the Hippo pathway effector Yorkie in C. owczarzaki (PMID: 35659869). This species is an amoeba that holds an important phylogenetic position as a close relative of multicellular animals. The prior study used genome editing to delete the C. owczarzaki Yki (termed coYki) and found that Yki is not required for proliferation but instead regulates cell contractility and cell aggregation. In the current study, the authors wanted to address whether Hippo pathway kinases - coHippo (coHpo) and coWarts (coWts) - regulate coYki and whether they are dispensable for proliferation but instead regulate cell contractility and cell aggregation. They used genome editing to delete coHpo and coWts singly in C. owczarzaki. Both mutant strains had increased nuclear location of transfected coYki (tagged with Scarlet), suggesting that Hippo kinase pathway regulation of Yki is conserved in this organism. Neither kinase is required for proliferation. Either kinase mutant strain had a significantly increased percentage of cells that were elongated, which was relatively rare in a control population. The incident of elongation could be enhanced in both kinase-mutant and in control cells when myosin inhibitors were added to the media. coHpo and coWts-mutant aggregates were more tightly packed than control cell aggregates, which they hypothesize is due to the increased contractility seen in kinase-mutant cells. They could reduce the density of packing in kinase-mutant aggregates when they treated the cells with myosin or F-actin inhibitors. To test whether the effects observed in kinase-mutant strains were due to increased Yki activation, they generated a coYki with four S to A substitutions (termed coYki4SA), which should produce a dominant-active Yki impervious to phosphorylation and hence inactivation by Hippo kinases. Control C. owczarzaki cells transfected with coYki4SA had increased cell density in aggregates and elongation in adherent cells. These results support their conclusions that coHpo and coWts regulate cell contractility and cell packing through coYki.

Strengths:

The major strengths of the paper include high quality data, robust analyses of the data, and a well-written manuscript. The combination of genome editing in C. owczarzaki, transfection of C. owczarzaki, and time-lapse movies of adherent cells generally support the conclusions (1) that control of cell density is an ancient function of the Hippo pathway; (2) that Hippo pathway control of cytoskeletal properties and contractile behavior underlie its regulation of cell density, and (3) that Hippo kinase control of Yki localization is likely an ancient function of the pathway.

Weaknesses:

There are no weaknesses.

---

## [Referee Report · Reviewer #2 (Public Review)]

The study builds on the work of the Pan group and others which has described the existence of core Hippo pathway proteins in Capsaspora and, more recently, described a role for a Yorkie/YAP homologue in regulation of cell shape and actin, as opposed to proliferation. For this recent study, they developed genetic techniques to mutate genes in Capsaspora, and this technology has been leveraged again in this study. Using loss of function genetic approaches, the authors find that loss of either of the two major kinases in the Hippo pathway core kinase cassette (Warts and Hippo) impact Capsaspora morphology and the actin cytoskeleton. This is phenocopied by overexpression of Capsaspora Yorkie/YAP. In addition, Capsaspora Yorkie/YAP accumulates in the nucleus of organisms lacking Warts or Hippo, as it does in metazoans. While these experiments are not overly surprising, they still provide important verification that core Hippo signaling events are conserved in Capsaspora.

Subsequently, they show that Capsaspora lacking Warts or Hippo do not overproliferate, which contrasts with many studies in metazoans (flies, mice, fish), particularly in epithelial tissues where loss of Warts or Hippo often causes overproliferation. Rather, the authors show that Capsaspora Warts and Hippo regulate cell morphology and actomyosin-dependent contractile behaviour. They speculate from these findings that Hippo signalling could regulate the density of Capsaspora when they grow in aggregates and draw parallels to the known role of the Hippo pathway in contact inhibition of mammalian cells grown in culture.

Together with their 2022 paper, this study paints an emerging picture that the ancestral function of the Hippo pathway is to regulate the actin cytoskeleton, not proliferation, which is a significant finding. This also suggests that the ability to control proliferation was something that the Hippo pathway was re-purposed to do at some stage during the evolution of metazoans. These findings are important for the Hippo field, and our understanding of cellular signalling and evolution more broadly.

In future studies, further biochemical and genetic experiments would allow the authors to more conclusively prove that core features of Hippo signalling are conserved in Capsaspora - e.g., that Capsaspora Hippo/MST activates Warts/LATS by phosphorylation and Warts/LATS represses Yorkie/YAP by phosphorylation hey serine residues. Some of these experiments are challenging or not yet possible due to technical limitations. Higher resolution imaging approaches such as electron microscopy would likely give further mechanistic insights into how Hpo, Wts and Yki modulate actomyosin contractility in Capsaspora. Recent advances in mass spectrometry of the phospho-proteome should provide a valuable way to explore Hippo signalling in Capsaspora. The benefit of this approach is it has the potential to give information on all Hippo pathway proteins and could be used to probe signalling events under different culture conditions (e.g., aggregate, non-aggregate).

---

## [Referee Report · Reviewer #3 (Public Review)]

The authors present in this study the characterization of two mutant lines of the filasterean Capsaspora owczarzaki, a unicellular holozoan with a key phylogenetic position to understand multicellular development in animals. The present study is built on a previous work from the same research group, on the mutant of the orthologue of the Yorki gene in C. owczarzaki. By knocking out the two upstream kinases of the same pathway, coHpo-/- and coWts-/-, in single cell and aggregates of C. owkzarzaki, they now have mutated the entire pathway and in different cellular contexts.

The authors obtain results in the same direction as the previous work, demonstrating that the Hippo pathway of the unicellular holozoan C. owczarzaki, is not involved in the control of cell proliferation but is related with cytoskeletal dynamics through the actin-myosin mechanism.

In this revised version of the study, the authors have addressed my concerns by providing additional experiments, references and discussing further the points of controversy.

I think the authors have done a great job improving the robustness of the paper proving further some of the claims raised in the previous version of the manuscript.

---

## [Author Response]

The following is the authors’ response to the original reviews.

**Reviewer #1 (Public Review):**
(1) Fig. 3C needs the "still" for the movie of control C. owczarzaki (in Movie S1).

We have now added a WT control in this figure panel.

(2) The elongated cell shape is seen infrequently in control cells, and I wonder whether these events are transient inactivation of coHpo or coWts in these cells. Perhaps the authors could comment on this in the discussion.

This is an interesting possibility and we have now included it in our discussion (Lines 401403).

(3) Does C. owczarzaki normally aggregate or this is a lab-specific phenotype? For example, the slime mold *Dictyostelium* discoideum forms aggregates during its life cycle. Could some additional information about C. owczarzaki be added to the introduction?

Unfortunately little is known about Capsaspora “in the wild”, as it was isolated as an endosymbiont from a laboratory strain of snails. However, some related filasterians isolated from natural environments also show aggregatve ability, indicating that aggregation is in fact a physiological process in this group of organisms. We have updated our introduction to include this fact (Line 78-80).

**Reviewer #2 (Recommendations For The Authors):**
The studies on Hippo signalling in Capsaspora are currently limited to genetic experiments and analysis of Yki/YAP localisation. Biochemical evidence that Co Wts phosphorylates Co Yki/YAP on a conserved serine residue(s) would give important further evidence that this essential signalling step in the animal Hippo pathway is conserved in Capsaspora. However, such experiments require antibodies that detect specific phosphorylation events, which might not be available at present. Is mass spectrometry of the phospho-proteome a potential approach that could be employed to investigate this? The benefit of this approach is it would give information on other Hippo pathway proteins and could be used to probe signalling events under different culture conditions (e.g., aggregate, non-aggregate).

In response to this recommendation, we attempted to detect Phospho-coWts and PhosphocoHpo using commercial antibodies against mammalian their homologs, in the hope of cross-species reactivity. However, we could not detect a signal by Western blot. Thus better reagents or refinement of techniques beyond the scope of this article may be required to examine the phosphorylation of these Capsaspora proteins. There was a published report of Capsaspora phosphoproteome analysis (Sebe Pedros et al., 2016 Dev Cell), although phosphorylation of the conserved sites on coYki, coWts, and coHpo was not reported in this analysis, suggesting more targeted approaches may be needed to examine phosphorylation of these core Hippo pathway components.

The following statement that Wts LOF is stronger than Hpo LOF Capsaspora is consistent with overgrowth phenotypes in flies and mammals:"Interestingly, we found that coWts-/- cells were significantly more likely to show nuclear mScarlet-coYki localization than coHpo-/- cells (Figure 1D), which is consistent with Hpo/MST independent activity of Wts/LATS previously reported in *Drosophila* and mammals (Zheng et al., 2015)."However, the following statement describes a stronger phenotype in Hpo LOF Capsaspora than Wts LOF:"As contractile cells in the coHpo mutant background tended to show a more extreme elongated morphology than the coWts mutant, we focused on the coHpo mutant for further analysis."Does this mean that Hpo can regulate actomyosin contractility in both Wts/Yki-dependent and independent manners? A genetic experiment, similar to those that have been performed in *Drosophila* and mammals could help to address this, e.g., what is the phenotype of Hpo, Yki Capsaspora and Wts, Yki double mutant Capsaspora? Do they phenocopy Yki LOF Capsaspora and are the actomyosin phenotypes associated with Hpo and Wts mutant Capsaspora completely or partially suppressed? The authors indicate that generation of double mutant Capsaspora is not technically possible at present, however.

Indeed given available techniques the generation of such double mutants is not currently possible. With this phenotype (aberrant cytoskeletal dynamics), it is hard to say what a “stronger” phenotype is, and which mutant has the “stronger” phenotype. We have edited this statement to try and reflect this point (Line 208-209).

Another outstanding question is whether the Hpo/Wts/Yki-related actomyosin phenotypes are linked to regulation of transcription by Yki, or are regulated non-transcriptionally. Indeed, a non-transcriptional role for *Drosophila* Yki in promoting actomyosin contractility has been reported (Fehon lab, Dev Cell, 2018). Generation of Scalloped/TEAD mutant Capsaspora would allow this question to be investigated. Alternatively, this could be explored using variant Co Yki transgenes, e.g., one a Co Yki transgene does not form a physical complex with Co Sd/TEAD and a Co Yki transgene that is targeted to the cell cortex.

To address this point, we tested whether a conserved amino acid residue in coYki (F123) that is required for transcriptional activity of human YAP (in this case, F95) is required for the phenotypic effects of the coYki 4SA mutant. We found that, in contrast to expression of coYki 4SA, expression of a coYki 4SA F123A mutant showed no effect on cell or aggregate morphology. These new results, which support a requirement for transcriptional activity for coYki function, have now been added to Figure 7.

**Reviewer #3 (Recommendations For The Authors):**
Repetition from previous publication:(1) ej: last sentences of the abstract in both works: From Phillips et al. eLife 2022;0:e77598: "Taken together, these findings implicate an ancestral role for the Hippo pathway in cytoskeletal dynamics and multicellular morphogenesis predating the origin of animal multicellularity, which was co-opted during evolution to regulate cell proliferation".From this manuscript: "Together, these results implicate cytoskeletal regulation but not proliferation as an ancestral function of the Hippo pathway and uncover a novel role for Hippo signaling in regulating cell density in a proliferation-independent manner "

Our two papers deal with different components of the Hippo pathway: Yorkie/YAP/coYki in Phillips et al. eLife 2022;0:e77598 and upstream kinases in the current paper. The fact that perturbing different components of the pathway leads to similar conclusions actually strengthens the overall conclusion. Nevertheless, to be more clear about the novelty of the current manuscript, we have now changed the current text from “Hippo pathway” to “Hippo kinase cascade”, to emphasize that the current analysis deals with kinases upstream of Yorkie/YAP/coYki (Lines 35, 368-371).

(2) The authors claim that the change in localization of coYki in Hpo -/- and Wts -/- , being now able to enter the nucleus, is the demonstration that the nuclear regulation of Yki by the Hippo pathway is ancestral to animals. Nevertheless, the authors had already made this claim in their publication of eLife 2022, when they made a mutant version of Yki with the four conserved phosphorylation sites (Sebé-Padrós 2012) mutated. Figure 5 A to F in Phillips et al. eLife 2022;0:e77598. In their words "This regulation of coYki nuclear localization, along with the previous finding that coYki can induce the expression of Hippo pathway genes when expressed in *Drosophila* (Sebé-Pedrós et al., 2012), suggests that the function of coYki has a transcriptional regulator and Hippo pathway effector is conserved between Capsaspora and animals. ".I understand that the localization of Yki in the coHpo-/- and coWts-/- is needed as part of final proof that Hpo and Wts are the kinases that control Yki phosphorylation in C. owczarzaki, but does not constitute a completely new message and should be written like that. Figure 1C of the actual manuscript drives to the same conclusion as Figure 5 A to F in Phillips et al. eLife 2022;0:e77598

We think that demonstrating that Hippo and Warts orthologs specifically are responsible for regulation of coYki localization is a very important finding: Many unicellular organisms encode Hippo, Warts, and/or Yorkie’s transcriptional factor partner Sd, but not Yorkie. Our understanding is that in these earlier-branching unicellular organisms, the Hippo/Warts kinase module and Sd-like proteins functioned in distinct signaling modules. Thus Yorkie has the interesting property of “fusing” these two distinct signaling modules when it emerged. In this framework, it is interesting to show that this “fusion” occurred in Capsaspora, the most distant known relative of animals with a Yorkie ortholog, indicating that this “fusion” event is very ancient. Although fleshing out of this idea is beyond the scope of this manuscript and we plan to write about it elsewhere, we have modified our discussion to point out the importance that Hippo and Warts specifically are upstream regulators of coYki.

In *Drosophila* among the genes transcriptionally regulated by Yki, are the positive regulators of the Hippo pathway in order to down regulate the Yki production.(1) The authors don't explain if these upstream regulators of the Hippo pathway are conserved in C. owczarzaki.

We have now indicated the conservation of some upstream Hippo pathway components (Line 69-71).

(2) Also it would be important to know how much coYki is being active in the C. owczarzaki in the mutant lines of coHpo-/- and coWts-/- in respect to wt and also in respect to coYki 4SA, and how this is impacting the transcription and protein production of down stream genes of coYki. I think some transcriptional and proteomic data would be informative. At least for those genes related with cytoskeleton.

We have now performed RNA-seq on the coHpo and coWts mutants to address the concerns above (See Figure 8 and the final section of Results).

Related with the above. Among the downstream targets of coYki, the authors mentioned in their previous work (Phillips et al. eLife 2022;0:e77598) that B-integrins were up regulated in coYki -/- suggesting that B-integrins could be behind the stronger cell-substrate attachment observed in the coYki-/- mutant. It would be important to investigate if the integrin adhesome is now down regulated and how previous and new results are related to the stronger cellsubstrate attachment in the coHpo-/- and coWts-/- lines. It would be important that previous results on coYki-/-, a mutant line of the same pathway, are discussed in these two new mutant contexts.

Two Capsaspora integrin beta genes were previously found to be upregulated in the coYki mutant (CAOG_05058 and CAOG_01283, from Phillips et al., 2022 eLife). In our coWts and coHpo mutant RNAseq data, we see that CAOG_05058 is upregulated in both coHpo and coWts mutants, whereas CAOG_01283 does not show significantly different expression in either the coHpo or coWts mutant. Because the CAOG_05058 expression data seems to go in the “opposite” direction than you might expect (i.e. not “down regulated” as the reviewer predicts), and because we see no change in expression in CAOG_01283, these results are difficult to interpret. Therefore the role of integrins in Capsaspora Hippo pathway mutant phenotypes is thus still an open question.

Some cells from the coHpo-/- and coWts-/- mutant lines, show higher attachment to the substrate, which results in an elongated shape while the cell detaches from the substrate. The authors claim this phenotype as a contractile behavior in these cells. This behavior would be caused by changes in cytoskeleton regulation or increased number of microvilli or a change in the distribution of microvilli.(1) In my opinion, this phenotype can not be considered a behavior per se (the cells become round once they are free from the substrate, so the elongation is temporal and the contractile behavior is a consequence from this attachment to the substrate), so I would not say that the Hippo pathway controls a contractile behavior as the authors state as one of the main conclusions of the manuscript.

Many cell behaviors are known to depend on external conditions, such as substrates, growth factors, nutrients, chemokines, etc., and are therefore “temporal” by the reviewer’s criteria. We therefore feel that the phenotype we describe here can be considered a cell behavior.

(2) On the other hand I think that further efforts on microscopy or immunocytochemistry could be performed in order to discern among the different causes; more microvilli? change in microvilli distribution? change in the acto-myosin cytoskeleton? Moreover these options are not mutually exclusive and very likely the explanation is multifactorial.(3) coWts-/- has a different phenotype at the periphery of the aggregates than coHpo-/-. The authors use stable transfected lines with NMM-Venus to visualize microvilli. It would be interesting that further experiments using this tool would be performed in order to visualize putative differences of the cell membrane at the periphery in the two mutant genotypes.

We have now performed experiments examining filopodia in round vs elongated cells using the NMM-venus marker, as well as differences in filopodial morphology within aggregates in the different genotypes. Our data and conclusions are included in our updated manuscript (Figure 3- figure supplement 1).

The authors nicely inspect the consequences of the mutant lines coHpo-/- and coWts-/- in the formation of the aggregates. They find that the aggregates in these cases are more densely packed likely due to the higher attachment from microvilli, which they are able to revert by using myosin inhibitors.(1) As mentioned above, it would be interesting that further experiments are performed by using NMM-Venus transfection into the coHpo-/- and coWts-/-genotypes in order to visualize putative differences of the strength and distribution of the microvilli in the aggregates of these two mutant genotypes. These experiments would inform if more or less microvilli contacts are created in these lines and support a mechanical explanation of the denser aggregates in the mutant lines, as they now suggest in the discussion.

We have now performed these experiments, and our data and conclusions are described in the updated manuscript (Figure 5- figure supplement 1).

(2) On the other hand, myosin inhibition through blebbistatin increases the number of elongated cells in the mutant lines, demonstrating that myosin is necessary for the cells to resolve their substrate attachment and become round. In my view is confusing that myosin is needed for cells to become round again (wt phenotype) and at the same time myosin inhibition is needed for aggregates to become less dense (wt phenotype). Do they lose density because more elongated cells are now in the aggregate? These results look confusing to me and I think they should be better discussed. Again the above transfections of NMM-Venus into the coHpo-/- and coWts-/-genotypes could be informative.

We have attempted to detect cells with an “elongated” morphology within WT and mutant aggregates but so far have been unable to visualize such cells. More advanced microscopy techniques at extended time scales may allow us to observe such things, but we believe such studies are beyond the scope of this manuscript.

The authors do not connect and discuss their results with a very relevant study done in *Drosophila*, Xu J et al. Dev Cell. 2018; 46(3): 271-284.e5, where a transcriptionally independent role of Yki is characterized. In *Drosophila*, Yki has an important role in a positive feedback loop with myosin at the cortical part of the cell, which is especially relevant for cytoskeleton regulation.The results encountered by the authors in their previous study with coYki-/-, indicated that coYki was important for proper actin dynamics and cell shape in C. owczarzaki. At that moment they did not interrogate if this phenotype could be due to the lack of a possible role of coYki in the cortex and they argue that the phenotype was caused by the lack of transcription regulation of downstream genes of coYki, which actually many were cytoskeleton related.Because the cortex function of Yki is independent of regulation of Hpo and Wts, the authors could use these genotypes by comparing them with WT (where the cortical role of Yki should be the same) and coYki-/- to investigate if the cortex role of Yki, is conserved in C. owczarzaki.In *Drosophila* the cortex role of Yki has been suggested to control tension at the cell surface. *Drosophila* Yki at the cortex activates myosin II through the N-terminal part of the protein and establishes a positive feedback loop by down regulating the Hippo pathway and obtaining therefore more active DmYki into the nucleus. This mechanism has been proposed by Xu et al.to work as the link between sensing cell tensions at the surface with control of tissue proliferation.In my opinion these are relevant results in the field that should be addressed in this study or at least well discussed. Actually, I think they could be a great opportunity for investigating if a putative cortex role of Yki is ancestral to its role linked to the Hippo pathway.

We have now addressed this study in our manuscript- please see our response to reviewer #2’s last comment above.

It would be informative to understand how stable expression through hygromicin selection is achieved in the transfection experiments. Is the recombinant plasmid integrated in the genome? Or is it stable as an episome?

We believe that the plasmids stably integrate, as we never lose fluorescent signal once established in a clonal line, even after extended culturing (>6 months). It may be worthwhile to definitely determine integration vs. episome in future studies.

The authors do not speculate or discuss how cell tension and cell proliferation is different for a unicellular organism or a tissue (multicellular) and I think should be addressed since the contexts are different.

This is an interesting and important point, which we plan to discuss in detail in an upcoming review article, as a proper discussion of this idea, we think, is beyond the scope of this manuscript.

Minor point. The study should cite other unicellular holozoans that have been also developed into treatable organisms such as Monosiga brevicollis (Woznica A, Kumar A, et al 2021eLife 10:e70436) and Abeoforma whisleri (Faktorová, D., Nisbet, R.E.R., Fernández Robledo, J.A. et al. Nat Methods17, 481-494 (2020)) in line 83 of the manuscript. I am sure the authors appreciate how much effort there is behind every non-model organism put forward as experimentally treatable and should be properly acknowledged.

We agree, and we have now included these examples of non-model organism development in our manuscript.